# FOXO1 is a master regulator of memory programming in CAR T cells

Alexander E. Doan[1,19], Katherine P. Mueller[2,3,4,19], Andy Y. Chen[5,6,7,19], Geoffrey T. Rouin[2,3,4], Yingshi Chen[2,3,4], Bence Daniel[5,7,8,9,18], John Lattin[1], Martina Markovska[2,3,4], Brett Mozarsky[2,3,4], Jose Arias-Umana[2,3,4], Robert Hapke[2,3,4], In-Young Jung[10], Alice Wang[2], Peng Xu[1], Dorota Klysz[1], Gabrielle Zuern[2,3,4], Malek Bashti[1], Patrick J. Quinn[1], Zhuang Miao[9], Katalin Sandor[5,7], Wenxi Zhang[5,7], Gregory M. Chen[4,11], Faith Ryu[2,3,4], Meghan Logun[4,12], Junior Hall[2,13,14], Kai Tan[2,13,14], Stephan A. Grupp[2,13,14], Susan E. McClory[2,13], Caleb A. Lareau[5,7,15], Joseph A. Fraietta[4,10,11,14], Elena Sotillo[1], Ansuman T. Satpathy[5,7,15], Crystal L. Mackall[1,15,16,17,20 ✉] & Evan W. Weber[2,3,4,13,14,15,20 ✉]

A major limitation of chimeric antigen receptor (CAR) T cell therapies is the poor persistence of these cells in vivo[1]. The expression of memory-associated genes in CAR T cells is linked to their long-term persistence in patients and clinical efficacy[2–6], suggesting that memory programs may underpin durable CAR T cell function. Here we show that the transcription factor FOXO1 is responsible for promoting memory and restraining exhaustion in human CAR T cells. Pharmacological inhibition or gene editing of endogenous *FOXO1* diminished the expression of memory-associated genes, promoted an exhaustion-like phenotype and impaired the antitumour activity of CAR T cells. Overexpression of FOXO1 induced a gene-expression program consistent with T cell memory and increased chromatin accessibility at FOXO1-binding motifs. CAR T cells that overexpressed FOXO1 retained their function, memory potential and metabolic fitness in settings of chronic stimulation, and exhibited enhanced persistence and tumour control in vivo. By contrast, overexpression of TCF1 (encoded by *TCF7*) did not enforce canonical memory programs or enhance the potency of CAR T cells. Notably, FOXO1 activity correlated with positive clinical outcomes of patients treated with CAR T cells or tumour-infiltrating lymphocytes, underscoring the clinical relevance of FOXO1 in cancer immunotherapy. Our results show that overexpressing FOXO1 can increase the antitumour activity of human CAR T cells, and highlight memory reprogramming as a broadly applicable approach for optimizing therapeutic T cell states.

More than 50% of patients who respond to CAR T cell therapies eventually relapse, and CAR T cells that target solid tumours have been largely ineffective[1]. The expression of memory T cell genes in patient CAR T cells is associated with durable persistence and disease control[2–6], but the transcription factors that drive CAR T memory programs have not been identified. We previously showed[7] that providing rest to exhausted CAR T cells through transiently inhibiting CAR signalling promoted a memory-like phenotype and increased chromatin accessibility at motifs bound by the memory transcription factors TCF1 and FOXO1, raising the prospect that these transcription factors mediate memory programming in CAR T cells. Consistent with this notion, expression of *TCF7* (which encodes TCF1) broadly correlates with responses to CAR T cell[2,5], tumour-infiltrating lymphocyte (TIL)[8] and checkpoint blockade[9,10] therapies. In addition, FOXO1 directly regulates the expression of *TCF7* and other canonical memory genes[11,12] and promotes the formation of central memory T cells in mice[12–14].

Several groups have shown that pharmacological inhibition of AKT, a negative regulator of FOXO1, confers an early memory phenotype in human CAR T cells and TILs[15–17], suggesting that FOXO1 also promotes memory in human T cells. To test the hypothesis that FOXO1 is required for memory programming and antitumour function in human CAR T cells, we performed phenotypic and functional experiments using

[1]Center for Cancer Cell Therapy, Stanford Cancer Institute, Stanford University School of Medicine, Stanford, CA, USA. [2]Department of Pediatrics, Perelman School of Medicine, University of Pennsylvania, Philadelphia, PA, USA. [3]Center for Cellular and Molecular Therapeutics, Children's Hospital of Philadelphia, Philadelphia, PA, USA. [4]Center for Cellular Immunotherapies, Perelman School of Medicine, University of Pennsylvania, Philadelphia, PA, USA. [5]Department of Pathology, Stanford University, Stanford, CA, USA. [6]Department of Bioengineering, Stanford University, Stanford, CA, USA. [7]Gladstone–UCSF Institute of Genomic Immunology, San Francisco, CA, USA. [8]Center for Personal Dynamic Regulomes, Stanford University, Stanford, CA, USA. [9]Department of Genetics, Stanford University, Stanford, CA, USA. [10]Department of Microbiology, Perelman School of Medicine, University of Pennsylvania, Philadelphia, PA, USA. [11]Department of Pathology and Laboratory Medicine, Perelman School of Medicine, University of Pennsylvania, Philadelphia, PA, USA. [12]Department of Neurosurgery, Perelman School of Medicine, University of Pennsylvania, Philadelphia, PA, USA. [13]Center for Childhood Cancer Research, Children's Hospital of Philadelphia, Philadelphia, PA, USA. [14]Abramson Cancer Center, Perelman School of Medicine, University of Pennsylvania, Philadelphia, PA, USA. [15]Parker Institute for Cancer Immunotherapy, San Francisco, CA, USA. [16]Department of Pediatrics, Stanford University, Stanford, CA, USA. [17]Department of Medicine, Stanford University, Stanford, CA, USA. [18]Present address: Genentech, South San Francisco, CA, USA. [19]These authors contributed equally: Alexander E. Doan, Katherine P. Mueller, Andy Y. Chen. [20]These authors jointly supervised this work: Crystal L. Mackall, Evan W. Weber. ✉e-mail: cmackall@stanford.edu; weberew@chop.edu

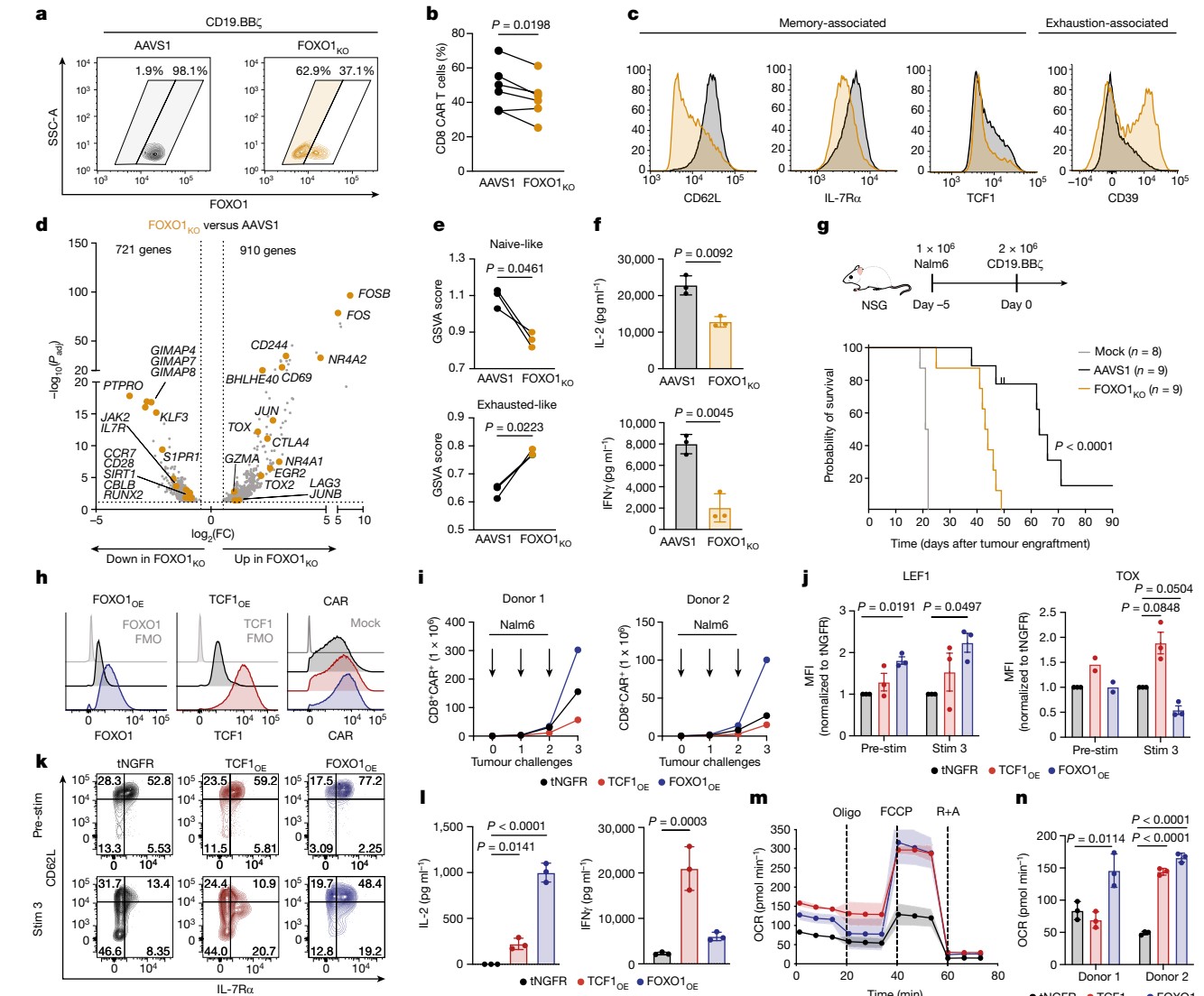

**Fig. 1 | FOXO1 is necessary and sufficient for memory and antitumour function in human CAR T cells. a–g**, CRISPR–Cas9 gene editing of *AAVS1* (AAVS1) or *FOXO1* (FOXO1_KO) in CD19.BBζ CAR T cells. **a–c**, Flow cytometric analysis of FOXO1 knockout efficiency (**a**), percentage of CAR+ CD8+ cells at day 14 (**b**) and memory- and exhaustion-associated markers in CAR+CD8+ cells (**c**). Shaded areas in **a** represent gates used in phenotypic analyses. One representative donor is shown in **a** and **c** (*n* = 6 donors). **d**, Volcano plot of DEGs in CD62L_lo FOXO1_KO versus AAVS1 (Bonferroni-adjusted *P* < 0.05 with absolute log_2-transformed fold change (abs(log_2(FC)) > 0.5). **e**, GSVA using T cell gene signatures[55]. **f**, Cytokine secretion in response to Nalm6 leukaemia cells from one representative donor (*n* = 4 donors). **g**, Stress test Nalm6 xenograft model. Top, schematic. Bottom, survival curves of Nalm6-engrafted mice treated with mock T cells or gene-edited CD19.BBζ cells. Data show two donors tested in two independent experiments (*n* = 8 or 9 mice per group). Data in **d** and **e** include *n* = 3 donors. **h–n**, CAR T cells overexpressing truncated NGFR (tNGFR), TCF1-P2A-tNGFR (TCF1_OE) or FOXO1-P2A-tNGFR (FOXO1_OE). **h**, Flow cytometric

analysis of FOXO1, TCF1 and CD19.28ζ expression from one representative donor (*n* = 8 donors). FMO, fluorescence minus one. **i–k**, Serial restimulation of CD19.BBζ cells with Nalm6. CD8+ CAR T cell expansion (**i**) and flow cytometric analysis of memory- and exhaustion-associated markers (**j,k**). **j**, Mean ± s.e.m. of normalized mean fluorescence intensity (MFI) (*n* = 2 or 3 donors). **k**, One representative donor (*n* = 4 donors). **l**, HA.28ζ cytokine secretion (day 13) in response to 143B osteosarcoma cells from one representative donor (*n* = 4 donors). **m,n**, HA.28ζ seahorse analysis (day 13) (*n* = 2 donors). **m**, Oxygen consumption rate (OCR) (mean ± s.d. of 11 technical replicates from one representative donor). Oligo, oligomycin; R+A, rotenone and antimycin. **n**, Spare respiratory capacity. Data in **f,l,n** are mean ± s.d. of three technical replicates. Statistical comparisons were performed using paired two-tailed Student's *t*-test (**b,e**), two-sided Welch's *t*-test (**f**), log-rank Mantel–Cox test (**g**) and repeated-measures one-way ANOVA with Geisser–Greenhouse correction (**j**) or one-way ANOVA with Dunnett's test (**l,n**).

CD19.28ζ or CD19.BBζ CAR T cells cultured in the presence of a selective FOXO1 small-molecule inhibitor[18] (FOXO1_i) (Extended Data Fig. 1). FOXO1_i reduced the expansion and viability of CAR T cells, the frequency of CD8+ cells and the expression of memory-associated markers (CD62L, IL-7Rα and TCF1) in a dose-dependent manner, and concomitantly upregulated markers of short-lived effector or exhausted T cells (CD39, TIM-3 and TOX) (Extended Data Fig. 1b–e).

We corroborated these data by using CRISPR–Cas9 to knock out *FOXO1* (FOXO1_KO) (Fig. 1a and Extended Data Fig. 2a). FOXO1_KO CAR T cells

showed a similar reduction in expansion and CD8+ frequency, diminished memory-associated markers and increased exhaustion-associated markers as compared with *AAVS1*-edited control CAR T cells (Fig. 1b,c, Extended Data Fig. 2b–f and Supplementary Fig. 1). Because FOXO1_KO cells exhibited uniformly low CD62L surface expression, we used CD62L as a surrogate marker for *FOXO1* editing by magnetically purifying CD62L_lo FOXO1_KO cells for bulk RNA sequencing (RNA-seq) (Extended Data Fig. 2g,h). FOXO1_KO cells upregulated activation- and exhaustion-associated genes (*TOX*, *NR4A1*, *FOS* and *CD69*),

downregulated memory and FOXO1 target genes (*IL7R* and *CCR7*) and exhibited less naive-like and more exhausted gene-expression signatures (Fig. 1d,e and Extended Data Fig. 2i).

FOXO1$_i$ and FOXO1$_{KO}$ cells also exhibited attenuated killing and/or cytokine secretion after tumour challenge (Fig 1f and Extended Data Fig. 1f,g), consistent with a model in which FOXO1 restrains exhaustion and/or terminal differentiation in human T cells, similar to reports in mice[14,19–22]. We corroborated these results using an in vitro CAR T cell exhaustion model (HA.28ζ CAR), in which antigen-independent tonic CAR signalling induces features of exhaustion within approximately one week[7,23]. Knockout of *FOXO1* in HA.28ζ cells accelerated the manifestation of exhaustion markers and dysfunction (Extended Data Fig. 2j,k). We next modelled chronic antigen stimulation in vivo by infusing a sub-therapeutic dose of CD19.BBζ cells into leukaemia-bearing mice[7,24]. Knockout of *FOXO1* significantly reduced CAR T cell tumour control and survival (Fig. 1g). These observations show that endogenous FOXO1 promotes memory and is required for optimal antitumour function of CAR T cells.

## FOXO1 overexpression preserves a memory phenotype

Among the genes induced by FOXO1 is *TCF7*, which has been broadly implicated in memory programming, stemness and antitumor activity in human and mouse T cells[2,5,8,10,25–33]. Thus, we sought to determine whether the overexpression of FOXO1 and/or TCF1 could enhance the function of human CAR T cells. Human T cells were co-transduced with a retrovirus expressing a CAR and a second virus expressing truncated NGFR (tNGFR) as a control or a bicistronic vector containing tNGFR and either TCF1 (TCF1$_{OE}$) or FOXO1 (FOXO1$_{OE}$) (Extended Data Fig. 3a). This approach enabled high levels of transcription factor overexpression and equivalent CAR expression across conditions (Fig. 1h). Notably, CD19.BBζ cells expressing FOXO1$_{OE}$, but not TCF1$_{OE}$, exhibited increased baseline expression of memory-associated surface markers and transcription factors, including endogenous TCF1 (refs. 12,13) (Extended Data Fig. 3b,c).

TCF1$_{OE}$ and FOXO1$_{OE}$ cells that were serially rechallenged with Nalm6 leukaemia both exhibited enhanced cytokine secretion compared with controls (Extended Data Fig. 3d), but only FOXO1$_{OE}$ increased CD8 proliferation and memory marker expression while suppressing the levels of TOX (Fig. 1i–k and Extended Data Fig. 3e). By contrast, TCF1$_{OE}$ increased the expression of TOX and CD39 relative to tNGFR controls, consistent with a more exhausted or effector-like phenotype (Fig. 1j and Extended Data Fig. 3e). We corroborated these results in cells expressing the tonic signalling HA.28ζ CAR, in which both TCF1$_{OE}$ and FOXO1$_{OE}$ cells showed enhanced function, but only FOXO1$_{OE}$ promoted a memory-like surface phenotype (Fig. 1l and Extended Data Fig. 3f–h).

Because the metabolism of memory T cells favours oxidative phosphorylation (OXPHOS) relative to glycolysis, we used Seahorse to assess whether transcription factor overexpression induces memory-like metabolic profiles. FOXO1$_{OE}$ and TCF1$_{OE}$ showed increased OXPHOS and superior metabolic fitness compared with tNGFR controls. The degree of FOXO1$_{OE}$-mediated metabolic reprogramming was more marked in exhausted HA.28ζ cells (Fig. 1m,n) compared with those expressing CD19.28ζ (Extended Data Fig. 3i,j), consistent with the notion that FOXO1$_{OE}$ counteracts the exhaustion program.

## FOXO1$_{OE}$ promotes a memory-like gene signature

We hypothesized that FOXO1 and TCF1 induce disparate gene-expression programs because overexpression of each endowed CAR T cells with distinct cell-surface phenotypes and functionality (Fig. 1). Therefore, we performed bulk RNA-seq on purified CD4$^+$ or CD8$^+$ FOXO1$_{OE}$ and TCF1$_{OE}$ T cells expressing HA.28ζ to model settings of chronic antigen stimulation. Principal component analysis (PCA) showed that FOXO1$_{OE}$ and TCF1$_{OE}$ CAR T cells clustered separately from tNGFR and had a greater number of unique differentially expressed genes (DEGs) than shared genes (Fig. 2a,b and Extended Data Fig. 4a–c). PCA also showed that transcription factor overexpression was a stronger driver of differential gene expression than CD4$^+$ or CD8$^+$ cell identity (Extended Data Fig. 4b), confirming that FOXO1$_{OE}$ and TCF1$_{OE}$ promote divergent gene-expression programs in both subsets.

Gene set variation analysis (GSVA) showed that FOXO1$_{OE}$ promoted a naive-like and less terminally exhausted gene signature (Fig. 2c). Consistent with these data, HA.28ζ FOXO1$_{OE}$ cells upregulated genes associated with memory (*SELL*, *IL7R*, *LEF1* and *TCF7*) and downregulated those associated with exhaustion (*TOX*, *HAVCR2*, *ENTPD1* and *CD244*) (Fig. 2d and Extended Data Fig. 4d,e). Despite the fact that previous literature has implicated FOXO1 in regulatory T (T$_{reg}$) cell biology[34,35], FOXO1$_{OE}$ did not enforce a T$_{reg}$ gene signature (Extended Data Fig. 4f). Gene ontology (GO) and ingenuity pathway analysis (IPA) showed that FOXO1$_{OE}$ promoted autophagy, cellular catabolism and naive-associated transcription factor gene-expression networks (*TCF7* and *LEF1*) and diminished effector transcription factor networks (*ID2*, *PRDM1* and *TBX21*) (Fig. 2e,f and Extended Data Fig. 4g,h). By contrast, TCF1$_{OE}$ cells exhibited high expression of exhaustion-associated transcription factors of the NR4A family, a progenitor exhausted T (T$_{pex}$) cell-like gene signature (Fig. 2c), and were enriched in effector gene-expression pathways (for example, cell–cell adhesion, T cell activation and cytokine production) (Fig. 2d,f and Extended Data Fig. 4d,e,g). Similar results were obtained in CD19.28ζ cells (Extended Data Fig. 4i–k); however, FOXO1$_{OE}$ resulted in a greater number of DEGs in tonic signalling HA.28ζ CAR T cells compared with those expressing CD19.28ζ, indicating more marked transcriptional reprogramming by FOXO1 during chronic stimulation.

TCF1 and FOXO1 are considered pioneer factors owing to their ability to directly bind to condensed chromatin and recruit chromatin remodelling machinery[36,37]. To test whether TCF1$_{OE}$ and/or FOXO1$_{OE}$ induce chromatin remodelling, we performed a bulk assay for transposase-accessible chromatin with sequencing (ATAC-seq) in TCF1$_{OE}$ and FOXO1$_{OE}$ CAR T cells (Supplementary Fig. 2). PCA confirmed that both transcription factors promoted global changes to chromatin accessibility compared with tNGFR controls (Fig. 2g and Extended Data Fig. 5a). This effect was most evident in tonically signalling HA.28ζ cells, in which FOXO1$_{OE}$ clustered separately from tNGFR and TCF1 groups and showed more differentially accessible peaks (around 5,600; $P < 0.05$) compared with TCF1$_{OE}$ cells (around 3,000) (Fig. 2g,h). Most of the differentially accessible peaks in FOXO1$_{OE}$ were open, consistent with the ability of FOXO1 ability to perturb core histone–DNA contacts[37].

HA.28ζ FOXO1$_{OE}$ cells showed increased accessibility at FOXO1 target gene loci (*IL7R* and *KLF3*), reduced accessibility at exhaustion-associated loci (*TOX* and *FASLG*) and a decreased exhaustion-like epigenetic signature compared with tNGFR cells (Fig. 2i,j), consistent with transcriptomic data. Of note, DNA-binding motifs for transcription factors of the forkhead box and HMG-box families were the top-ranked differentially accessible motifs in FOXO1$_{OE}$ and TCF1$_{OE}$ cells, respectively (Fig. 2k,l and Extended Data Fig. 5b,c), supporting a model in which overexpressed FOXO1 and TCF1 induce local chromatin remodelling. Paradoxically, FOXO1$_{OE}$ cells also showed increased accessibility at transcription factor motifs associated with effector function (for example, b-ZIP and NF-κB p65) (Extended Data Fig. 5d,e).

These data show that FOXO1$_{OE}$ induces memory and naive-like gene-expression programs during chronic stimulation, whereas TCF1$_{OE}$ promotes a T$_{pex}$-like program, consistent with the role identified for TCF1 in chronic infection and cancer[25,26,38,39]. In addition, FOXO1$_{OE}$ induces a unique epigenetic state that supports effector function while maintaining memory programming.

## FOXO1$_{OE}$ enhances CAR T function against leukaemia

Because FOXO1$_{OE}$ was effective at promoting memory (Fig. 1), we hypothesized that further increasing the activity of FOXO1 might

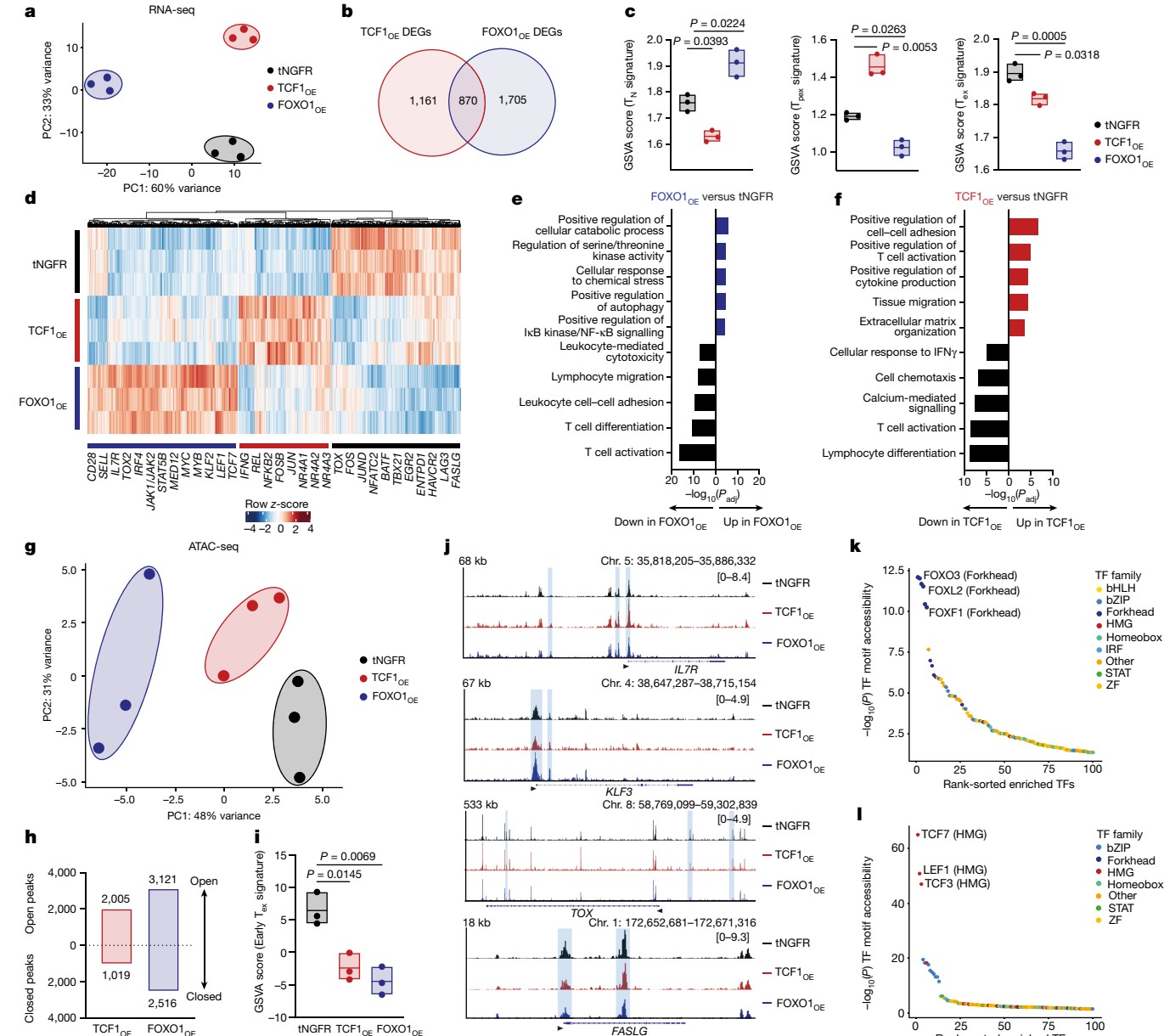

**Fig. 2 | Overexpression of FOXO1, but not TCF1, induces transcriptional and epigenetic features of T cell memory. a–l**, Bulk RNA-seq (**a–f**) and ATAC-seq (**g–l**) in tNGFR$^+$CD8$^+$ HA.28ζ CAR T cells ($n = 3$ donors). **a**, RNA-seq PCA. **b**, Venn diagram showing unique and shared DEGs in TCF1$_{OE}$ and FOXO1$_{OE}$ compared with tNGFR (Bonferroni-adjusted $P < 0.05$ with abs(log$_2$(FC) > 0.5). **c**, GSVA of DEGs using naive (T$_N$), T$_{pex}$ and exhausted (T$_{ex}$) T cell signatures[55]. Centre line represents mean score. **d**, Heat map and hierarchical clustering of DEGs. Genes of interest are highlighted. The colour bar shows normalized $z$-scores for each DEG. **e,f**, GO term analyses showing curated lists of the top upregulated and downregulated processes in FOXO1$_{OE}$ (**e**) and TCF1$_{OE}$ (**f**) versus tNGFR (Benjamini–Hochberg-adjusted $P$). **g**, ATAC-seq PCA. **h**, Number of differentially accessible peaks compared with tNGFR ($P < 0.05$ with abs(log$_2$FC) > 0.5). **i**, GSVA of differentially accessible peaks using an early T$_{ex}$ cell epigenetic signature[9]. Centre line represents mean score. **j**, Chromatin accessibility tracks for the *IL7R*, *KLF3*, *TOX* and *FASLG* loci, for one representative donor. **k,l**, Rank-ordered plots of differentially accessible transcription factor (TF)-binding motifs in FOXO1$_{OE}$ (**k**) and TCF1$_{OE}$ (**l**) versus tNGFR. ZF, zinc-finger. Statistical comparisons were performed using DESeq2 (**b,d,h,k,l**), one-sided hypergeometric test (**e,f**) and repeated-measures one-way ANOVA with Dunnett's test (**c,i**).

endow CAR T cells with a more stable memory phenotype. We generated a humanized version of a nuclear-restricted variant of FOXO1 (FOXO1$_{3A}$), which is insensitive to AKT-mediated nuclear export[19] (Extended Data Fig. 6a–c and Supplementary Fig. 3). FOXO1$_{3A}$ increased the surface expression of FOXO1 target genes to a similar extent to FOXO1$_{OE}$ (Extended Data Fig. 6d,e). However, FOXO1$_{3A}$ expression induced a divergent transcriptomic profile that was de-enriched in T cell activation genes and led to blunted in vitro cytokine secretion and cytotoxicity compared with FOXO1$_{OE}$ (Extended Data Fig. 6f–i). These observations raised the prospect that excessive nuclear FOXO1 activity

might promote a stable memory phenotype and oppose effector function[21].

To assess function in a protracted model in which memory programming might be important for sustained antitumor activity, we used a stress test xenograft model in which leukaemia-bearing mice received a sub-therapeutic dose of CD19.28ζ (Fig. 3a) or CD19.BBζ (Extended Data Fig. 7a) CAR T cells. FOXO1$_{OE}$ markedly enhanced the tumour control of CAR T cells compared with tNGFR, whereas TCF1$_{OE}$ showed no benefit (Fig. 3a and Extended Data Fig. 7a,b). Similar results were obtained in a curative Nalm6 model, in which FOXO1$_{OE}$ cells exhibited

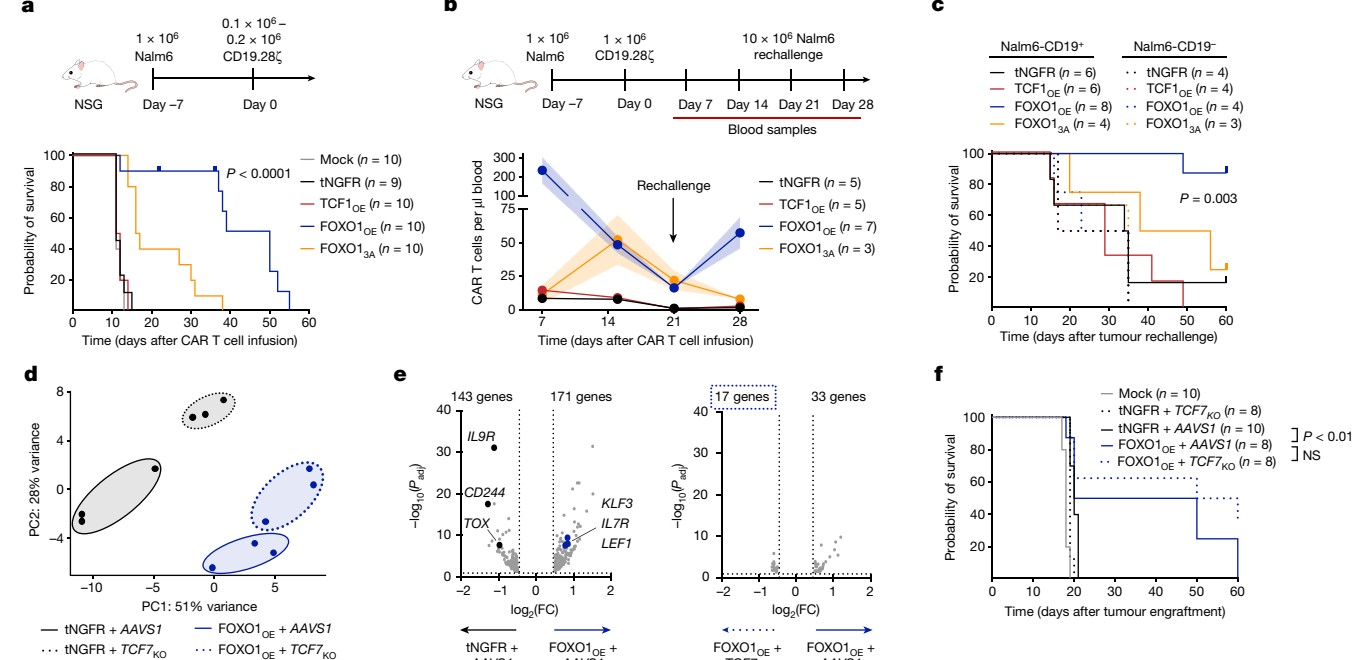

**Fig. 3 | Overexpression of FOXO1 enhances CAR T cell persistence and antitumour activity against leukaemia in a *TCF7*-independent manner.** **a**, Subcurative doses of $0.1 \times 10^6$–$0.2 \times 10^6$ tNGFR⁺ CD19.28ζ cells were infused into Nalm6-bearing mice seven days after engraftment. Schematic (top) and survival curve (bottom) are shown; $n$ = 9-10 mice per group. **b**–**d**, Curative doses of $1 \times 10^6$ tNGFR⁺ CD19.28ζ cells were infused into Nalm6-bearing mice seven days after engraftment. Mice were rechallenged with $10 \times 10^6$ CD19⁺ or CD19⁻ Nalm6 on day 21 after CAR T cell infusion ($n$ = 2 donors tested in 2 independent experiments). **b**, Rechallenge Nalm6 model. Schematic (top) and quantification (bottom) of circulating human CD45⁺ CAR T cells. Mean ± s.e.m. of $n$ = 3-7 mice

per group from one representative donor. **c**, Survival curve after rechallenge ($n$ = 3–8 mice per group pooled from 2 donors). **d**–**f**, CD19.28ζ cells overexpressing tNGFR or FOXO1_OE were gene-edited to knock out *AAVS1* (control; *AAVS1*) or *TCF7* (*TCF7*_KO). **d**, RNA-seq PCA. **e**, Volcano plots of DEGs; $n$ = 3 donors (Bonferroni-adjusted $P < 0.05$ with abs(logFC) > 0.5). **f**, Stress test Nalm6 model. tNGFR⁺ CD19.28ζ cells ($0.6 \times 10^6$ cells) were infused into Nalm6-bearing mice seven days after engraftment. Survival curve is shown ($n$ = 8–10 mice per group). **a**,**c**,**f** show pooled data from two donors tested in two independent experiments. Statistical comparisons were performed using log-rank Mantel–Cox test (**a**,**c**,**f**) and DESeq2 (**e**). NS, not significant.

increased expansion and persistence compared with TCF1_OE and tNGFR cells (Fig. 3b and Extended Data Fig. 7c–e). FOXO1_3A provided a modest survival advantage compared with tNGFR, but FOXO1_3A cells exhibited delayed expansion and reduced levels of tumour control compared with FOXO1_OE cells (Fig. 3a–c), consistent with the notion that FOXO1_3A partially opposes effector function. To assess the recall response to secondary antigen challenge—a hallmark feature of memory T cells[40]—we rechallenged nearly cured mice with a high dose of Nalm6 (Fig. 3b,c and Extended Data Fig. 7c–e). Only FOXO1_OE cells re-expanded after rechallenge and conferred a survival advantage, showing that FOXO1_OE endows CAR T cells with superior in vivo effector- and memory-like functions compared with tNGFR, TCF1_OE or FOXO1_3A.

### *TCF7* is not required for FOXO1_OE reprogramming

To investigate the mechanism by which FOXO1_OE reprograms CAR T cells and increases in vivo antitumour activity, we generated a variant of FOXO1 with lower-affinity DNA binding (FOXO1_DBD)[41]. FOXO1_DBD showed a modest reduction in DNA binding, and its expression in CAR T cells perturbed FOXO1-mediated transcriptional and epigenetic reprogramming (Extended Data Fig. 8a–d). Mice that received CD19.28ζ FOXO1_DBD cells showed reduced survival in a Nalm6 leukaemia stress test model compared to those that were infused with FOXO1_OE cells (Extended Data Fig. 8e), indicating that FOXO1_OE DNA binding is crucial for augmented antitumour activity.

The FOXO1 target gene, *TCF7*), is highly upregulated in FOXO1_OE cells (Extended Data Figs. 3c and 4d). Although TCF1_OE did not increase the potency of CAR T cells, we reasoned that high endogenous levels of *TCF7* and expression kinetics in FOXO1_OE could be mechanistically

important for FOXO1_OE reprogramming. Notably, knockout of *TCF7* in the context of FOXO1_OE had negligible effects on FOXO1_OE transcriptional reprogramming and in vivo antitumour activity (Fig. 3d–f and Extended Data Fig. 8f). Thus, FOXO1_OE reprogramming requires DNA binding but not transcription of the memory-associated transcription factor and target gene, *TCF7*.

### FOXO1_OE enhances CAR T function in solid tumours

To determine whether FOXO1 was also capable of increasing the activity of CAR T cells against solid tumours, we infused tNGFR or FOXO1_OE HER2.BBζ CAR T cells into 143B osteosarcoma-bearing NSG mice. Consistent with leukaemia models, FOXO1_OE cells showed durable antitumour activity and persistence (Fig. 4a–e and Extended Data Fig. 9a–d). Tumour-infiltrating FOXO1_OE cells exhibited transcriptomic reprogramming, were enriched in gene signatures associated with T cell killing, effector function and tissue residence, and showed negligible differences in human T_reg signatures[42,43] (Fig. 4f–h and Extended Data Fig. 9e–i). Of note, intratumoral FOXO1_OE cells did not have a canonical memory-like phenotype but were enriched in a FOXO1_OE transcriptomic signature derived from bulk RNA-seq studies (Fig. 4h), suggesting that exogenous FOXO1 remains active in the tumour microenvironment.

Together, these data show that FOXO1_OE increases the in vivo expansion, persistence and tumour control of CAR T cells in a *TCF7*-independent manner, whereas TCF1_OE provides no measurable benefit. FOXO1_OE-mediated enhancements are dependent on DNA binding and nuclear export, which suggests that tuning or signal regulation mediated by nuclear shuttling is important for effective FOXO1-mediated memory programming.

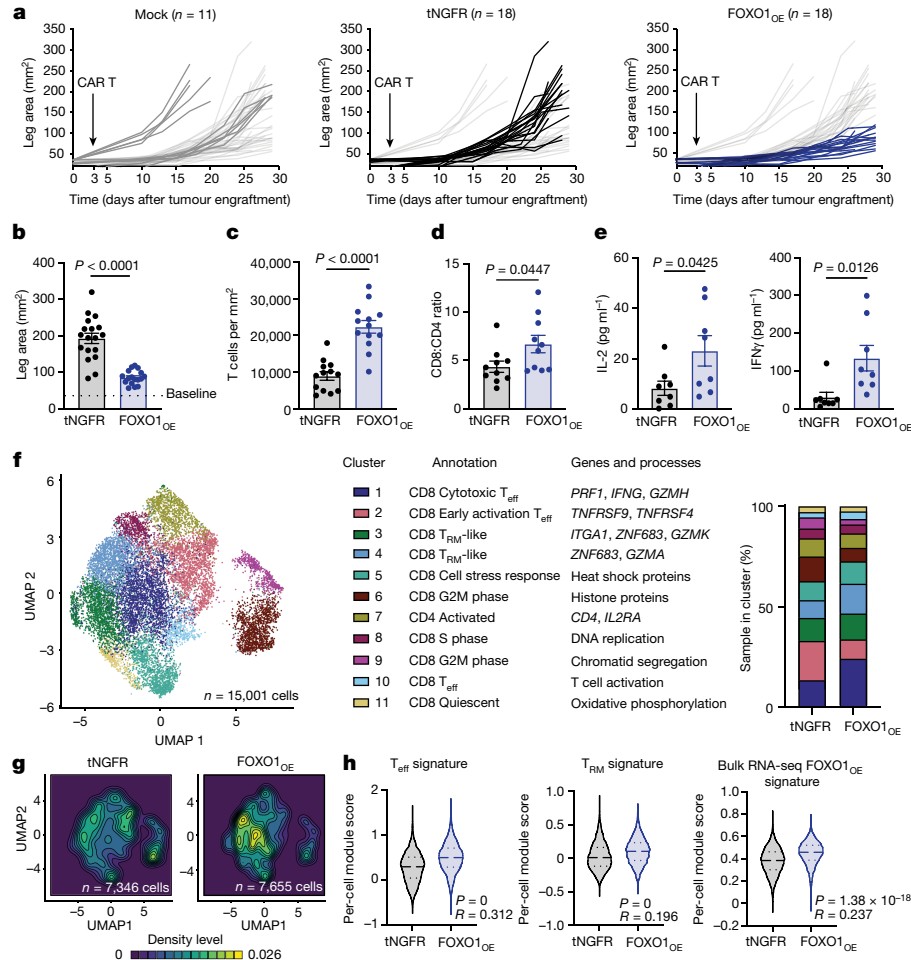

**Fig. 4 | FOXO1_OE CAR T cells exhibit enhanced tumour control and sustained effector function in solid tumours.** A total of $5 \times 10^6$ mock or tNGFR[+] Her2.BBζ CAR T cells expressing tNGFR or FOXO1_OE were infused into 143B-bearing mice three days after engraftment. **a**,**b**, Tumour measurements over time (**a**) and on day 25–29 (**b**). One FOXO1_OE mouse has been omitted in **b** owing to tumour-independent death before day 25. Data were pooled from three donors tested in three independent experiments ($n = 11$–18 mice per group). **c**–**e**, Analysis of day-29 CAR TILs. **c**, Total CAR TILs ($n = 13$ mice per group). **d**, Ratio of CD8[+] to CD4[+] CAR TILs. One representative donor ($n = 10$ mice per group). **e**, CAR TIL IL-2 and IFNγ secretion after ex vivo stimulation with 143B ($n = 13$ mice per group). Data in **c**–**e** were pooled from two donors tested in two independent experiments. **f**–**h**, Single-cell RNA-seq on day-29 CAR TILs. Cells were sorted and pooled from $n = 5$ mice per group from one donor. **f**, Left, uniform manifold approximation and projection (UMAP) of CAR TILs. Eleven clusters were identified with $k$-nearest neighbours clustering, and were annotated manually (middle). Right, sample distribution by cluster. T_eff, T effector cell; T_RM, tissue resident memory T cell. **g**, Sample distribution within the UMAP. **h**, T_eff, T_RM and FOXO1_OE-associated transcriptional signatures. Long dashed lines represent the mean and short dashed lines represent the top and bottom quartiles. Data in **b**–**e** are mean ± s.e.m. Statistical comparisons were performed using two-tailed Student's $t$-test (**b**–**e**) and two-sided Wilcoxon rank-sum test (**h**).

## FOXO1 activity correlates with response to T cell therapies

FOXO1 target genes, including *TCF7*, were enriched in pre-infusion CAR T cells that mediate clinical responses in patients[2,5] (Extended Data Fig. 10a,b), raising the possibility that endogenous FOXO1 activity might predict potent antitumour activity in clinical CAR T products. Paradoxically, however, *FOXO1* transcript levels in pre-infusion CD19. BBζ cells were not associated with response to therapy or survival in adults with chronic lymphocytic leukaemia (CLL) (Fig. 5a and Extended Data Fig. 10c). Because FOXO1 is regulated mainly post-translationally rather than transcriptionally[15], we hypothesized that the activity of FOXO1 could be better approximated by the aggregate expression of FOXO1 target genes. We therefore identified a FOXO1 'regulon' consisting of 41 overlapping DEGs that were downregulated in FOXO1_KO cells and upregulated in FOXO1_OE cells (Fig. 5b). The FOXO1 regulon included putative FOXO1 target genes (for example, *SELL* and *KLF3*), but was made up largely of genes that have not previously been associated with memory programming (Supplementary Table 1). In contrast to *FOXO1*

transcript, the FOXO1 regulon was significantly enriched in pre-infusion CAR T cells from patients with CLL who exhibited complete or partial responses with transformed disease, and was associated with in vivo CAR T cell expansion and overall survival (Fig. 5c,d and Extended Data Fig. 10d). *TCF7* did not reach statistical significance in FOXO1_KO experiments and was therefore not included in the FOXO1 regulon; however, regulon score significantly correlated with the *TCF7* transcript in patient CAR T cells, suggesting that the regulon is an accurate readout for FOXO1 transcriptional activity (Fig. 5e).

The FOXO1 regulon was also enriched in pre-manufactured effector T cells from children with B cell acute lymphoblastic leukaemia (B-ALL) who exhibited durable CAR T cell persistence[5] (Fig. 5f), supporting the notion that FOXO1 activity broadly correlates with the efficacy of CAR T cells. Because both FOXO1 and TCF1 mediate chromatin remodelling[36,37,44–46] (Fig. 2), we next used epigenetic signatures derived from our ATAC-seq analyses to interrogate single-cell ATAC-seq data from paediatric CAR T cells[5]. Consistent with FOXO1 regulon transcriptomic data, the FOXO1_OE epigenetic signature was significantly enriched in patient T cells that were associated with durable persistence, whereas

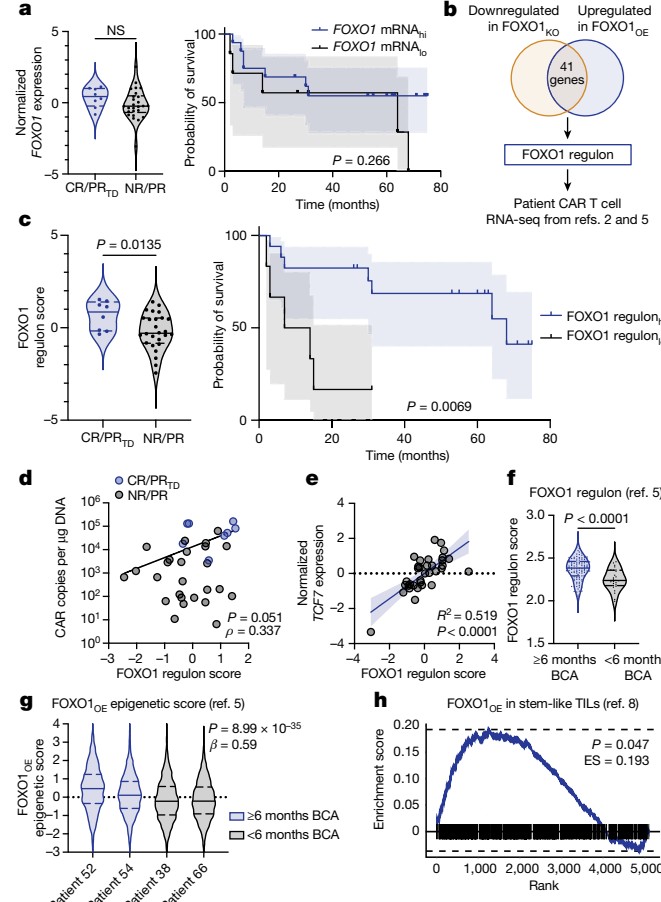

**Fig. 5 | FOXO1 activity correlates with clinical responses to CAR T cell and TIL therapies. a–e**, Single-sample gene set enrichment analysis (ssGSEA) on RNA-seq from pre-infusion, CAR-stimulated CTL019 cells from patients with CLL[2] (complete responder (CR), $n = 5$; partial responder with transformed disease (PR$_{TD}$), $n = 3$; partial responder (PR), $n = 5$; non-responder (NR), $n = 21$). **a**, *FOXO1* ssGSEA for patient outcomes (left) and overall survival (right). **b**, The FOXO1 regulon was generated using FOXO1$_{KO}$ and FOXO1$_{OE}$ bulk RNA-seq data and then applied to published datasets[2,5]; $n = 3$ donors. **c**, FOXO1 regulon ssGSEA (data from ref. 2) for patient outcomes (left) and overall survival (right). **d**, Least squares regression (dark line) of FOXO1 regulon score and peak CAR T cell expansion. **e**, Simple linear regression (dark line) of *TCF7* expression and FOXO1 regulon score. Dark lines in **a**,**c** represent patient survival curves and shaded areas in **a**,**c**,**e** represent 95% confidence intervals. Dots in **d**,**e** represent individual samples (blue, CR/PR$_{TD}$; grey, NR/PR). **f**, FOXO1 regulon ssGSEA for pre-manufactured effector T cells from paediatric patients with B-ALL with durable (six or more months of B cell aplasia (BCA); $n = 33$ patients) or short (less than six months of BCA; $n = 27$ patients) CAR T cell persistence[5]. **g**, An epigenetic signature derived from FOXO1$_{OE}$ ATAC-seq was applied to pre-manufactured T cell single-cell ATAC-seq data from paediatric patients[5]. Data show FOXO1$_{OE}$ epigenetic signature scores for patients with durable (patient 52, $n = 616$ cells; patient 54, $n = 2,959$ cells) and short (patient 38, $n = 2,093$ cells; patient 66, $n = 2,355$ cells) CAR T cell persistence. **h**, GSEA using FOXO1$_{OE}$ DEGs and DEGs derived from CD39$^-$CD69$^-$ TILs from adult patients with melanoma[8]. ES, enrichment score. Violin plots in **a**,**c**,**f**,**g** show minima and maxima; solid lines represent the mean and long dashed lines represent the top and bottom quartiles. Statistical comparisons were performed using two-tailed Mann–Whitney test (**a**, left; **c**, left; **f**), log-rank Mantel–Cox test (**a**, right; **c**, right), Spearman correlation (**d**,**e**), two-sided Wald test (**g**) and two-sided Kolmogorov–Smirnov test (**h**).

the TCF1$_{OE}$ signature was not (Fig. 5g and Extended Data Fig. 10e). Finally, FOXO1$_{OE}$ DEGs were enriched in stem-like CD39$^-$CD69$^-$ TILs that were highly predictive of the response to TIL therapy in adult patients

with melanoma[8], whereas TCF1$_{OE}$ DEGs were de-enriched (Fig. 5h and Extended Data Fig. 10f).

## Discussion

In this study, we tested the hypothesis that overexpressing memory-associated transcription factors could reprogram CAR T cells to durably persist and maintain antitumor activity. We focused our efforts on FOXO1, on the basis of studies that have implicated this transcription factor in memory programming[12–14,19–21,46–51] and our previous work in which we showed that exhaustion reversal and memory programming were associated with enhanced chromatin accessibility at FOXO1-binding motifs[7]. FOXO1 overexpression induced memory gene-expression programs and chromatin remodelling, mitigated exhaustion and substantially improved persistence and antitumour function in four distinct xenograft models. Its effect was independent of CAR binder, co-stimulatory domain and tumour type, highlighting the broad applicability of this pro-memory program across CAR T cell products.

There is a vast body of literature describing the role of FOXO1 in promoting T cell memory and persistence in mice[12–14,19–21,46–51]; however, FOXO1 biology in human T cells remains poorly understood. Because the activity of FOXO1 is regulated at the post-translational level rather than through changes in transcription and is therefore hidden in RNA-seq data, the role of FOXO1 in cancer immunology and immunotherapy is likely to have been considerably underappreciated. Our study is the first, to our knowledge, to show that endogenous FOXO1 is required for memory gene expression and optimal antitumour function in engineered human T cells, which is consistent with the effects of *Foxo1* knock-out in mouse models of acute and chronic infection[14,19,20]. We further show that endogenous FOXO1 restrains exhaustion in human T cells, because deleting *FOXO1* induced an exhaustion-like phenotype and CAR T cell dysfunction.

Notably, FOXO1 activity in pre-infusion CAR T cells and TILs strongly correlated with clinical responses, underscoring the importance of FOXO1 in T-cell-based cancer immunotherapies. Paradoxically, expression of a nuclear-restricted variant (FOXO1$_{3A}$) altered FOXO1 reprogramming and attenuated the antitumour function of CAR T cells, supporting the notion that optimal FOXO1 activity involves intermittent and/or context-dependent regulation. Indeed, others have shown that transient expression of FOXO1$_{3A}$ can induce partial memory reprogramming in human CAR T cells without impairing effector function[15,52,53]. Further work is needed to determine how FOXO1 expression levels and kinetics affect the function of CAR T cells and whether FOXO1 is relevant in other therapeutic modalities, such as immune checkpoint blockade.

We also interrogated TCF1, a transcription factor that defines stem-like or memory T cell populations that exhibit an increased capacity to respond to immune checkpoint blockade[2,5,8,10,25–33]. Of note, over-expressing TCF1 did not enforce memory gene-expression programs or enhance antitumour activity in vivo, which contradicts reports in mice[27,28]. Instead, TCF1$_{OE}$ cells exhibited a gene-expression signature associated with T$_{pex}$ cells, and manifested functional hallmarks of exhaustion during chronic stimulation, consistent with other studies[39,54]. Thus, our results raise the possibility that constitutive TCF1 overexpression skews human engineered T cells towards a more exhausted or T$_{pex}$ cell-like state, and/or that *TCF7*-expressing T$_{pex}$ cells do not have a substantial role in CAR T cell responses.

An alternative interpretation posits that FOXO1, rather than TCF1, is mainly responsible for endowing tumour-reactive T cells with a stem-like or progenitor phenotype, and that *TCF7* expression is merely a readout for FOXO1 activity. Indeed, deletion of endogenous *TCF7* in FOXO1$_{OE}$ did not affect FOXO1-mediated transcriptional reprogramming or augmented antitumour function in vivo. Surface markers and transcription factors that are often co-expressed in *TCF7*$^+$ cells are

FOXO1 target genes[29], and our empiric FOXO1 regulon significantly correlated with *TCF7* expression and clinical responses in samples of CAR T cells from patients, further supporting this notion. Conditional deletion of *Foxo1* in mature mouse T cells diminished the frequency of *Tcf7*-expressing $T_{pex}$ cells[14], suggesting that FOXO1 might promote cell states that are normally associated with high levels of *Tcf7* expression. Future mechanistic studies are warranted to determine the precise roles of FOXO1 and TCF1 in human engineered and non-engineered T cells during cancer immunotherapy.

In summary, we show that FOXO1-driven transcriptional and epigenetic programs are associated with engineered and non-engineered T cells that expand, persist and promote clinical responses in patients with cancer. Overexpression of FOXO1 increases the activity of CAR T cells through memory reprogramming, and TCF1 is insufficient to induce CAR T cell memory and persistence. Our results suggest that FOXO1 represents a major therapeutic axis that can be exploited to improve the efficacy of T-cell-based cancer immunotherapies.

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

## Methods

### Primary human T cells

For experiments completed at Stanford, buffy coats from anonymous, consenting healthy donors were obtained from the Stanford University Blood Center under an University Institutional Review Board-exempt protocol or obtained from a human peripheral blood leukopak (STEMCELL Technologies). CD3[+] cells were isolated using the RosetteSep Human T Cell Enrichment Kit, Lymphoprep density gradient medium and SepMate-50 tubes according to the manufacturer's protocol (STEMCELL Technologies). For experiments completed at the Children's Hospital of Philadelphia (CHOP), purified CD3[+] healthy donor T cells were obtained from the University of Pennsylvania Human Immunology Core. All purified T cells were cryopreserved in CryoStor CS10 medium (STEMCELL Technologies).

### Cell lines

Cell lines were obtained from ATCC and stably transduced to express markers as follows: 143B osteosarcoma cells express GFP and firefly luciferase with or without CD19, Nalm6 B-ALL cells express GFP and firefly luciferase with or without GD2. Single-cell clones were chosen for high antigen expression. The 143B and Nalm6 cells were cultured in Dulbecco's modified Eagle's medium (DMEM) and RPMI 1640, respectively, and both were supplemented with 10% fetal bovine serum (FBS), 10 mM HEPES and 1× penicillin–streptomycin–glutamate (Gibco). Nalm6 and 143B cell lines and engineered versions of these cell lines were previously authenticated via STR fingerprinting prior to their use in this study. HEK293 cells were originally obtained from the National Cancer Institute. Cells were frequently tested for mycoplasma using the Lonza MycoAlert Mycoplasma Detection kit.

### Design of CAR and transcription factor constructs

The CAR constructs used in this study include CD19.28ζ, CD19.BBζ, anti-GD2 HA.28ζ and Her2.BBζ. Codon-optimized TCF1, FOXO1 or FOXO1$_{3A}$ sequences and a P2A ribosomal skip sequence were generated as Gene Blocks by IDT and constructed in MSGV retroviral vectors. The tNGFR-only construct does not contain a P2A ribosomal skip sequence. The FOXO1$_{DBD}$ construct was generated by two-step mutagenic NEBuilder HiFi DNA Assembly (New England BioLabs). All plasmids were amplified by transformation into Stellar Competent *Escherichia coli* (Takara Bio), and sequences were validated by sequencing (Elim Biopharmaceuticals).

### Retrovirus production

To generate retrovirus, ten million 293GP cells were plated on a 15-cm BioCoat poly-D-lysine cell culture plate (Corning) and fed with 20 ml of DMEM supplemented with 10% FBS, 10 mM HEPES and 1× penicillin–streptomycin–glutamate (Gibco) 24 h before transfection. Transfection was performed by mixing a room-temperature solution of 3.4 ml Opti-MEM (Gibco) + 135 μl Lipofectamine 2000 (Invitrogen) (solution 1) with a second solution of 3.4 ml Opti-MEM + 11 μg RD114 packaging plasmid DNA + 22 μg MSGV retroviral plasmid of interest (solution 2) by slow dropwise addition of solution 2 to solution 1. The combined solution 1 and 2 mixture was incubated for 30 min at room temperature, after which the medium was replaced on 293GP cells, and 6.5 ml of the combined solution was added to the plates in a slow, dropwise manner. The next day, the culture medium was replaced on 293GP cells. At 48 h after transfection, the viral supernatant was collected from the cells and the culture medium was replaced; supernatant collection was repeated at 72 h. At each step, the supernatant was spun down to remove cells and debris, and frozen at −80 °C for future use.

### T cell activation and culture

T cells were thawed in warm water after removal from liquid nitrogen and then washed with T cell medium (AIM-V (Gibco) supplemented with 5% FBS, 10 mM HEPES, 1× penicillin–streptomycin–glutamate and 100 U ml$^{-1}$ recombinant human IL-2 (Peprotech) or RPMI (Gibco) supplemented with 10% FBS, 10 mM HEPES, 1× penicillin–streptomycin–glutamate and 100 U ml$^{-1}$ recombinant human IL-2). Human T-Expander αCD3/CD28 Dynabeads (Gibco) were washed and added to T cells at a volume of 30 μl resuspended beads per million T cells. T cells and beads were then resuspended at a concentration of 500,000 T cells per ml in T cell medium (day 0 for all assays). Forty-eight and 72 hours after activation, T cells were transduced (see 'Retroviral transduction'). Ninety-six hours after activation, beads were removed by magnetic separation using a DynaMag column (Invitrogen). T cells were fed with fresh T cell medium every 48–72 h and were maintained at a density of 0.5 ×10$^6$ cells per ml after feeding. For FOXO1$_i$ experiments, T cells were provided with fresh complete T cell medium and vehicle control (dimethyl sulfoxide; DMSO) or AS1842856 (EMD Millipore) every 2–3 days from days 4 to 15 after activation.

### Retroviral transduction

T cells were transduced with retrovirus on days 2 and 3 after activation for all experiments. In brief, 12- or 24-well, non-tissue-culture-treated plates were coated with 1 ml or 500 μl, respectively, of 25 μg ml$^{-1}$ Retronectin (Takara) in PBS and placed at 4 °C overnight. The next day, plates were washed with PBS then blocked with 2% bovine serum albumin (BSA) + PBS for 10 min. Retroviral supernatants were added and plates were centrifuged at 32 °C for 2 h at 2,500$g$. Viral supernatants were subsequently removed and T cells were added to each virus-coated well at a density of 1 × 10$^6$ T cells per well for 12-well plates and 0.5 × 10$^6$ T cells per well for 24-well plates.

### Cell selection

tNGFR isolations were performed using either Miltenyi MACS sorting or STEMCELL EasySep sorting unless otherwise stated. For Miltenyi MACS sorting, cells were resuspended in FACS buffer and stained with biotin anti-human CD271 (tNGFR) antibody (BioLegend). Cells were washed with PBS, 0.5% BSA and 2 mM EDTA (MACS buffer), resuspended in MACS buffer and mixed with Streptavidin MicroBeads (Miltenyi), then washed again with MACS buffer and passed through an LS Column for positive selection inside a MACS separator (Miltenyi). For STEMCELL EasySep sorting, cells were isolated using the manufacturer's protocol for the EasySep Human CD271 Positive Selection Kit II (STEMCELL Technologies) with an EasyEights EasySep Magnet (STEMCELL Technologies). After isolation, cells were immediately mixed with warm complete T cell medium, counted and resuspended at 500,000 per ml.

For RNA-seq experiments on FOXO1$_{KO}$ cells, CD62L$_{lo}$ CAR$^+$ cells were isolated by negative selection, first by staining cells with anti-CD62L-PE and then by following the EasySep PE Positive Selection Kit II protocol according to the manufacturer's instructions (STEMCELL Technologies). For RNA-seq and ATAC-seq experiments on tNGFR, TCF1$_{OE}$ and FOXO1$_{OE}$ cells, CD8$^+$tNGFR$^+$ CAR T cells were isolated before sequencing using the EasySep Human CD8+ T Cell Isolation Kit (STEMCELL Technologies). For in vivo analysis of tumour-infiltrating CAR T cells, CD45$^+$ T cells were isolated from tumours using the EasySep Release Human CD45 Positive Selection Kit (STEMCELL Technologies) according to the manufacturer's instructions.

### CRISPR–Cas9 gene editing

To interrogate the role of endogenous FOXO1 in CAR T cell function, CRISPR–Cas9 was used to delete a sequence directly upstream of the *FOXO1* DNA-binding domain. On day 4 after activation, retrovirally transduced CAR T cells were removed from activation beads by magnetic separation. Twenty-microlitre reactions were prepared by resuspending one million CAR T cells in P3 buffer immediately before electroporation with the P3 Primary Cell 4D Nucleofector Kit (Lonza). Ribonucleoproteins were prepared by complexing 0.15 ng of sgRNA targeting *FOXO1* or *AAVS1* (Synthego) with 5 μg Alt-R S.p. Cas9 Nuclease (IDT, 1081058)

before adding the cell suspension to each reaction. For *AAVS1* edits, a previously validated sgRNA sequence (5′-GGGGCCACUAGGGACAG GAU-3′) was used. For *FOXO1*, two separate sgRNAs were used in tandem, at equal concentrations (5′-UUGCGCGGCUGCCCCGCGAG-3′ and 5′-GAGCUUGCUGGAGGAGAGCG-3′). For *TCF7* gene editing, we used a previously validated sgRNA[56] (5′-UCAGGGAGUAGAAGCCAGAG-3′) for bulk RNA-seq experiments performed at CHOP. A separate sgRNA (5′-UUUUCCAGGCCUGAAGGCCC-3′) was designed and validated at Stanford, and used for in vivo experiments. The reaction was pulsed with the EH115 program on a Lonza 4D Nucleofector. Cells were recovered immediately in 260 μl of warm complete AIM-V medium supplemented with 500 U ml$^{-1}$ IL-2 in round-bottom 96-well plates and expanded into 1 ml fresh medium within 24 h. Cells were maintained at $0.5 \times 10^6$ cells per ml to $1.0 \times 10^6$ cells per ml in well plates until day 14–16 for functional and phenotypic characterization. On days 14–16, knockout efficiency was determined by intracellular transcription factor staining (Cell Signaling, 58223) followed by flow cytometry.

## Flow cytometry
CAR T cells were washed twice in FACS buffer (PBS + 2% FBS) and stained with fluorophore-conjugated surface antibodies for 30 min on ice. Cells were washed twice with FACS buffer before analysis. Intracellular stains were performed with the same initial surface stain, after which cells were fixed, permeabilized and stained using the FoxP3 Transcription Factor Staining Buffer Set according to the manufacturer's protocol (eBioscience). Anti-human FOXO1 (clone C29H4) and anti-human TCF1 (C36D9) antibodies were purchased from Cell Signaling. The 1A7 anti-14G2a idiotype antibody used to detect the HA CAR was obtained from the NCI and conjugated using the Dylight 650 antibody labelling kit (Thermo Fisher Scientific). The anti-FMC63 idiotype antibody was manufactured by GenScript and fluorescently conjugated using the Dylight 650 antibody labelling kit. Cell-surface antibodies were used at a 1:100 dilution during staining, with the exception of anti-14g2a and anti-FMC63, which were used at a 1:1,000 dilution. Intracellular antibodies were used at a 1:50 dilution and live/dead staining was used at a 1:1,000 dilution. Cells were analysed with either a BD Fortessa running FACS Diva software, or a Cytek Aurora using SpectroFlo v.3.1.0. Downstream analyses were performed using Cytek SpectroFlo v.3.1.0 and FlowJo v.10.8.1 Software. All reagents are listed in Supplementary Table 2. A representative gating strategy for FOXO1$_{KO}$ and FOXO1$_{OE}$ experiments is shown in Supplementary Fig. 1. In experiments in which we stained for Annexin V, cells were gated on all singlets, excluding debris but not excluding dead or dying T cells. For MFI quantification, background subtraction was performed using either unstained or FMO samples. The MFI quantification in Extended Data Fig. 1e was not background subtracted owing to negative MFI values in some control samples.

## Cytokine secretion assays
A total of $5 \times 10^4$ CAR T cells were co-cultured with $5 \times 10^4$ tumour cells in 200 μl of complete T cell medium (AIM-V or RPMI) without IL-2 in a 96-well plate, all in triplicate. Twenty-four hours after co-culture, culture supernatants were collected, diluted 20- to 100-fold and analysed for IL-2 and IFNγ using ELISA MAX kits (BioLegend) and Nunc Maxisorp 96-well ELISA plates (Thermo Fisher Scientific). Absorbance readings were collected on a Tecan Spark plate reader or a BioTek Synergy H1 running Gen5 v.2.00.18. For FOXO1$_i$ assays, the co-culture medium included concentrations of AS1842856 that were used during T cell expansion.

## IncuCyte killing assay
A total of $5 \times 10^4$ GFP$^+$ tumour cells and T cells corresponding to a 1:1, 1:2, 1:4, 1:8 and/or 1:16 effector:target ratios were co-cultured in 300 μl of T cell medium without IL-2 in 96-well flat-bottom plates. Plates were imaged at 10× zoom with 4–9 images per well every 2–4 h for 96 h using

the IncuCyte ZOOM S3 Live-Cell analysis system (Essen BioScience/Sartorius). The total integrated GFP intensity per well or total GFP area (μm$^2$ per well) were used to analyse the expansion or contraction of Nalm6 or 143B cells, respectively. All GFP intensity and area values were normalized to the first imaging time point ($t = 0$). For FOXO1$_i$ assays, the co-culture medium included concentrations of AS1842856 that were used during T cell expansion.

## Repeat stimulation assay
CAR T cells were activated and transduced, and tNGFR$^+$ cells were isolated as described above. Cells were cultured in AIM-V with IL-2 until day-14 'pre-stim' assays, including flow cytometry, cytokine secretion and IncuCyte as described above. On day 14, co-cultures were set up comprising $5 \times 10$ T cells and $2 \times 10^6$ Nalm6 tumour cells suspended in AIM-V without IL-2 at a final concentration of $5 \times 10^5$ total cells per ml. Co-cultures were fed with 5 ml of AIM-V without IL-2 on day 3 of culture. On day 3 of the repeat stimulation co-culture, CAR T cells were again assayed by cytokine secretion, IncuCyte killing assay and flow cytometry as described above. This process was repeated for a total of four co-cultures such that the cytokine and IncuCyte assays were set up for four serial stimulations on days 14, 17, 20 and 23 on cells that had been stimulated with Nalm6 tumour zero, one, two and three previous times, respectively, for a total of four serial stimulations by the end of the experiment. Cells were analysed by flow cytometry on day 7 of co-culture, such that T cells were co-cultured with tumour on days 14, 17, 20 and 23 and analysed on days 21, 24, 27 and 30, respectively.

## Seahorse assay
Metabolic analyses were performed using Seahorse Bioscience Analyzer XFe96. In brief, $0.2 \times 10^6$ cells were resuspended in extracellular flux assay medium supplemented with 11 mM glucose, 2 mM glutamine and 1 mM sodium pyruvate, and plated on a Cell-Tak (Corning)-coated microplate allowing the adhesion of CAR T cells. Mitochondrial activity and glycolytic parameters were measured by the oxygen consumption rate (OCR) (pmol min$^{-1}$) and extracellular acidification rate (ECAR) (mpH min$^{-1}$), respectively, with the use of real-time injections of oligomycin (1.5 M), carbonyl cyanide ptrifluoromethoxyphenylhydrazone (FCCP; 0.5 M) and rotenone and antimycin (both at 0.5 M). Respiratory parameters were calculated according to the manufacturer's instructions (Seahorse Bioscience). Reagent sources are listed in Supplementary Table 2.

## Immunoblotting
Chromatin-bound and soluble proteins were separated as previously described[23]. In brief, cytoskeletal (CSK) buffer was prepared using 100 mM NaCl, 300 mM sucrose, 3 mM MgCl$_2$, 10 mM PIPES (pH 6.8), 0.1% IGEPAL CA-630, 4 μg ml$^{-1}$ aprotinin, 10 μg ml$^{-1}$ leupeptin, 4 μg ml$^{-1}$ pepstatin and 2 mM PMSF. After washing with ice-cold PBS, cell pellets were lysed with CSK buffer for 20 min on ice. Samples were centrifuged at 1,500$g$ for 5 min and the soluble fraction was separated and cleared by centrifugation at 15,870$g$ for 10 min. The protein concentration of the soluble fraction was determined by DC protein assay (Bio-Rad, 5000116). The remaining pellet containing the chromatin-bound fraction was washed twice with CSK buffer, centrifuging at 1,500$g$ for 5 min. Chromatin-bound proteins were resuspended in CSK buffer and 1× Pierce Reducing Sample Buffer (Thermo Fisher Scientific, 39000) and boiled for 5 min for solubilization. The soluble fraction was supplemented with Pierce Reducing Sample Buffer to achieve 1× and boiled for 5 min. For immunoblotting, equal amounts of soluble and chromatin-bound fraction for each sample were analysed by SDS−polyacrylamide gel electrophoresis and transferred to nitrocellulose membranes (Bio-Rad, 1704158). Membranes were blocked for 30 min in 5% milk in TBST (1× Tris-buffered saline containing 0.1% Tween-20). After washing with TBST, membranes were incubated with anti-FOXO1 antibody (1:1,000; Cell Signaling, 2880, clone C29H4)

overnight at 4 °C. Next, membranes were washed with TBST and incubated with anti-mouse (1:10,000, Cell Signaling, 7074) or anti-rabbit (1:10,000, Cell Signaling, 7076) IgG conjugated to horseradish peroxidase for 1 h at room temperature. Membranes were visualized using Clarity Western ECL Substrate (Bio-Rad, 1705060) and the ChemiDoc Imaging System and Image Lab Touch Software v.3.0 (Bio-Rad). After visualization, membranes were stripped using a mild stripping buffer (1.5% glycine, 0.1% SDS, 1% Tween-20, pH 2.2). The previous steps were repeated for detection of soluble (1:5,000 GAPDH; Cell Signaling, 97166, clone D4C6R) and chromatin-bound (1:1,000 Lamin A; Cell Signaling, 86846, clone 133A2) fraction loading controls. Densitometry analyses were performed using Fiji v.2.14.0/1.5 f.

### Mouse xenograft models

NOD/SCID/*Il2rg*$^{-/-}$ (NSG) mice were bred, housed and treated under Stanford University APLAC- or CHOP ACUP-approved protocols. Six-to-eight-week-old mice were healthy, immunocompromised, drug- and test-naive and unused in other procedures. Mice were housed at the Stanford Veterinary Service Center (VSC) or CHOP Department of Veterinary Services (DVR) in a barrier facility with a 12-h light–dark cycle, and mice were kept at a temperature of 20–23 °C (CHOP) or 20–26 °C (Stanford) with humidity ranging from 30–70%. Five mice were housed in each cage in aerated racks with ample bedding, food and water. For mice that became sick, solid feeds were switched to liquid feeds to facilitate eating. Mice were monitored daily by trained VSC and DVR staff under the supervision of a veterinarian who reported excess morbidity immediately and/or euthanized mice for humane reasons. Mice were euthanized if end-point criteria were met, which included 143B tumour sizes exceeding 1.2 cm or Nalm6 bioluminescence greater than $5 \times 10^{11}$ photons per second, or if evidence of extensive disease occurred (for example, inability to ambulate, groom or eat, cachexia, excessive loss of fur, hunched posture or other signs of disability); whichever came first. Tumour injection sites were chosen so as not to interfere with the mouse's normal body functions, such as ambulation, eating, drinking, defecation and/or urination. In Nalm6-bearing mice, $2 \times 10^5$ to $1 \times 10^7$ cells in 100–200 μl of sterile PBS were engrafted by tail vein injection (TVI). In 143B osteosarcoma models, $1 \times 10^6$ to $3 \times 10^6$ cells in 100 μl sterile PBS were engrafted by intramuscular injection into the flank. Mice were randomized prior to CAR T cell infusion to ensure equal tumour burden across groups. CAR T cells were engrafted by TVI at doses and schedules noted in the main text. Nalm6 engraftment, expansion and clearance were measured by intraperitoneal injection of luciferin and subsequent imaging by a Spectrum IVIS bioluminescence imager and quantified using Living Image software v.4.7.3 (Perkin Elmer), or by a Lago X imager and quantified using Aura software v.4.0.7 (Spectral Instruments Imaging), all under isoflurane anaesthesia. The 143B tumour size was monitored by caliper measurements. Tumor and T cell injections were performed by technicians who were blinded to treatments and expected outcomes.

### Mouse tissue analyses

Peripheral blood was sampled from live, isoflurane-anaesthetized mice by retro-orbital blood collection. Fifty microlitres of blood was labelled with surface antibodies, lysed using FACS Lysing Solution (BD) and quantified using CountBright Absolute Counting Beads (Thermo Fisher Scientific), then analysed on a BD Fortessa cytometer. For phenotypic analysis of spleen and tumours, mice were euthanized and tissues were mechanically dissociated and washed twice in PBS. Spleens were placed in a 6-cm Petri dish and filtered through a sterile 70-μm cell strainer. Tumours were mechanically and chemically dissociated with Collagenase IV and DNAse in HBSS and incubated at 37 °C with shaking for 30 min. Cells were mashed through a sterile 70-μm cell strainer before washing with PBS. Cells from both spleens and tumours were spun down at 450*g* for 5 min at 4 °C, then treated with ACK lysis buffer for 3 min on ice. Cell suspensions were washed

twice with PBS and CAR T cells were isolated by positive selection using the EasySep Release Human CD45 Positive Selection Kit. Cells were stained for markers of interest and analysed on a Cytek Aurora using SpectroFlo Software 3.1.0.

### Bulk RNA-seq

A total of $0.5 \times 10^6$–$1 \times 10^6$ T cells were pelleted by centrifugation and flash-frozen. Pellets were thawed on ice and processed using either an RNEasy Plus Mini Kit or an AllPrep DNA/RNA Micro Kit (for simultaneous DNA and RNA isolation) (QIAGEN) according to the manufacturer's instructions. Total RNA was quantified using either a Qubit Fluorometer or a DeNovix DS-11 FX Spectrophotometer/Fluorometer and sequenced using a 150 bp paired-end read length and around 50 million read pairs per sample (Novogene).

### Bulk RNA-seq processing and analysis

We processed the sequencing data using the nf-core RNA-seq pipeline (https://nf-co.re/rnaseq). In brief, we performed quality control of the fastq files using FastQC and trimmed the filtered reads with Trim Galore software. The trimmed fastq files resulting from the experiment were aligned to the hg38 human genome using STAR. Salmon was then used to generate a gene-by-sample count matrix for downstream analysis. PCA was performed on read counts that were processed using the variance-stabilizing transformation, and plots were generated from the top 1,000 variable genes across samples. To correct for batch effects by donor, the removeBatchEffect function in the limma package was used. Differential analysis of gene expression was performed using the DESeq2 v.3.16 package, with an absolute $\log_2$-transformed fold change ≥0.5 and false discovery rate (FDR) < 0.05. To create a heat map, differential genes were aggregated, and expressions were standardized with *z*-scores across samples. The *k*-means clustering algorithm with Pearson correlation as the distance metric was used to cluster the genes. Pathway analysis of the differential genes and grouped genes in the heat map was performed using QIAGEN Ingenuity Pathway Analysis 2022 Winter Release and clusterProfiler v.4.6.2. Cell-type enrichment was performed through the single-sample extension of gene set enrichment analysis (ssGSEA) in the GSVA v.1.46.0 R package using signature genes from previous studies[8,55] using R v.4.1.0.

### Single-cell RNA-seq library preparation and sequencing

To generate single-cell RNA-seq libraries of tumour-infiltrating CAR T cells, Her2$^+$ tumours were collected from five mice per condition, and human CD45$^+$ cells were isolated by NGFR selection as described above (see 'Cell selection'). Tumour-infiltrating CAR T cells were further purified by sorting human CD3$^+$ TILs from each isolate using a Cytek Aurora Cell Sorter. A total of 20,000 CAR TILs were sorted from each tumour and pooled across five mice per group. Cells were barcoded and sequencing libraries were generated using the 10X Chromium Next GEM Single Cell 3' v.3.1 kit (10X Genomics) according to the manufacturer's instructions. Libraries were sequenced at the CHOP High Throughput Sequencing Core on an Illumina NovaSeq 6000 with an average read depth of 50,000 reads per cell.

### Single-cell RNA-seq processing and analysis

FASTQ files were generated and aligned to the genome with Cell Ranger v.7.1.0, using a custom GRCh38 reference genome containing the Her2. BBζ CAR sequence. Low-quality cells with fewer than 300 or more than 7,500 genes or more than 10% mitochondrial reads were removed using Seurat v.4.3.0 (ref. 57) in R. Doublets were identified using Doublet-Finder v.2.0.3 and removed. Filtered samples were normalized using SCTransform before integration. The integrated dataset was scaled, and UMAP dimensionality reduction was performed using the top 30 principal components. Unsupervised Louvain clustering was performed on a shared nearest neighbour graph at a final resolution of 0.6. FindAllMarkers (Seurat) was used to identify DEGs in each cluster,

and GO analyses were performed for each cluster using ClusterProfiler v.4.6.2. DEGs and GO processes were used to manually annotate each cluster, and contaminating CD3⁻ tumour cells were removed. Differential gene analyses between samples were performed using FindMarkers (Seurat) using the Wilcoxon rank-sum test with Bonferroni correction. Gene set scores for $T_{eff}$, $T_{RM}$ and $T_{reg}$ cell subtypes were calculated with AddModuleScore (Seurat), using curated gene lists from a previous study[58] (Extended Data Fig. 9g–i). AddModuleScore was also used to calculate a per-cell FOXO1 transcriptional activity score, using the top 100 upregulated genes in CD8⁺ HA.28ζ CAR T cells overexpressing FOXO1 versus tNGFR (Fig. 2). Gene set scores for $T_{eff}$, $T_{RM}$ and FOXO1 signatures were generated for pan CD3⁺ T cells (Fig. 4i; individual genes are shown in Extended Data Fig. 9g–i). The $T_{reg}$ gene set score was computed for the CD4⁺ subset of cells expressing ≥1 *CD4* mRNA counts and no detectable *CD8A* counts (Extended Data Fig. 9f).

### Bulk ATAC-seq processing

CD8⁺tNGFR⁺ CAR T cells were isolated using the EasySep Human CD8+ T Cell Isolation Kit. A total of 150,000 CD8⁺ T cells were slow-frozen in BamBanker (Bulldog Bio) cell preservation medium. Approximately 100,000 CAR T cells were washed in ice-cold PBS and subjected to nuclei isolation using the following lysis buffer: 10 mM Tris-HCl pH 7.5, 10 mM NaCl, 3 mM MgCl₂, 0.1% Tween-20, 0.1% NP40, 0.01% Digitonin and 1% BSA. After washing the cells, 50 µl lysis buffer was added to each sample and cells were resuspended by pipetting. Nuclear pellets were centrifuged and resuspended in the transposase reaction containing 10.5 µl H₂O, 12.5 µl 2× TD buffer and 2 µl Tn5 transposase in a total of 25 µl. The reaction was incubated for 30 min at 37 °C. The reaction was stopped by the addition of 75 µl TE buffer and 500 µl PB buffer (QIAGEN), followed by column purification per the manufacturer's recommendation (QIAGEN, Minelute Kit). DNA was eluted from the columns in 22 µl H₂O. PCR reactions were set up as follows: 21 µl DNA, 25 µl Phusion master mix (NEB) and 2 µl of each barcoded PCR primer (ApexBio, K1058). Fifteen PCR cycles were run for each sample. Reactions were cleaned up with AMPure XP beads according to the recommendations of the manufacturer. Libraries were quantified with a Qubit fluorometer and fragment analysis was performed with Bioanalyzer. Libraries were sequenced on a NovaSeq 6000 sequencer.

### Bulk ATAC-seq analysis

ATAC-seq libraries were processed using the pepatac pipeline (http://pepatac.databio.org/) with default options. In brief, fastq files were trimmed to remove adapter sequences, and then pre-aligned to the mitochondrial genome to exclude mitochondrial reads. To ensure the accuracy of downstream analysis, multimapping reads aligning to repetitive regions of the genome were filtered from the dataset. Bowtie2 was then used to align the reads to the hg38 genome. SAMtools was used to identify uniquely aligned reads, and Picard was used to remove duplicate reads. The resulting deduplicated and aligned BAM file was used for downstream analysis. Peaks in individual samples were identified using MACS2 and compiled into a non-overlapping 500-bp consensus peak set. In brief, the peaks were resized to 500 bp width and ranked by significance. The peaks that overlapped with the same region were selected by ranks and the most significant peak was retained. The peak-sample count matrix was generated using ChrAccR with the default parameters of the run_atac function. Signal tracks for individual samples were generated within the pepatac pipeline. These tracks were then merged by group using WiggleTools to produce a comprehensive view of the data across all samples.

On the basis of our analysis of the peak-sample count matrix, the DESeq2 v.3.16 package was used to identify differential peaks across different conditions, with a threshold of an absolute log₂-transformed fold change greater than 0.5 and *P* value less than 0.05. Adjusted *P* values were not used owing to donor variability. To generate PCA plots, we first extracted a variance-stabilized count matrix using the vst function in DESeq2. Next, we corrected for batch effects by donor using the removeBatchEffect function in the limma library. Finally, we generated PCA plots using the corrected matrix with the plotPCA function using the top 2,000 most variable peaks. We aggregated differential peaks across conditions, standardized the peak signals using *z*-scores across samples and performed *k*-means clustering to generate a chromatin accessibility heat map. Motif enrichments of differential peaks and grouped peaks were searched with HOMER and findMotifsGenome.pl with default parameters. The enrichment of cell-type-specific regulatory elements were performed with the gchromVAR package. In brief, this method weights chromatin features by log₂-transformed fold changes of cell-type-specific regulatory elements from a previous report[9] and computes the enrichment for each cell type versus an empirical background matched for GC content and feature intensity.

### Identification and analysis of the FOXO1 regulon

The FOXO1 regulon gene set was generated by intersecting downregulated differential genes (log₂-transformed fold change < −0.25, FDR < 0.05) in $FOXO1_{KO}$ cells and upregulated differential genes (log₂-transformed fold change > 0.5, FDR < 0.05) in $FOXO1_{OE}$ cells (Supplementary Table 1). Regulon enrichment scores were calculated using ssGSEA in the GSVA R package on a previous RNA expression dataset[2].

For regulon analyses of single-cell ATAC-seq data, the processed Signac data objects of CAR T products profiled by single-cell ATAC-seq were obtained from a previous study[5]. To account for sample-to-sample variability, the mean fragments in peaks per cell were downsampled for consistency between donors. Furthermore, donors PT48 and PT51 were excluded on the basis of low data quality after examination of quality control statistics, including per-library transcription start site enrichment. Using the epigenetic signature for FOXO1 and TCF1 overexpression (Fig. 2), we computed the per-cell epigenetic signature per factor using the chromVAR workflow as previously described for related T cell signatures derived from bulk experiments. To test for differences in responder/non-responder associations with this signature, we performed an ordinary least squares regression with the per-cell *z*-score against the donor's BCA status at 6 months, adjusting for individual patient ID. Statistical significance was based on the Wald test statistic of the coefficient for the responder term in the two regressions for each factor.

For regulon analyses of the CLL CD19 CAR T cell clinical dataset, the gene-expression data table for activated CD19 CAR T cell products from patients with CLL was obtained from a previous report[2]. The enrichment of the FOXO1 signature was analysed using ssGSEA as previously described and performed using the R package GSVA v.1.46.0. To compare the ssGSEA enrichment scores between responders and non-responders, a Mann–Whitney test was conducted. To statistically determine optimal stratification points for survival analysis, we compared candidate stratification points on the basis of hazard ratio and *P* value as previously described. The survival analysis was conducted with a log-rank (Mantel–Cox) test using GraphPad Prism v.9.5.0.

### Statistical analyses

Unless otherwise stated, statistical analyses for significant differences between groups were conducted using one- or two-way analysis of variance (ANOVA) with Bonferroni, Tukey's or Dunnett's multiple comparisons test, or with a Student's or Welch's *t*-test using GraphPad Prism v.9.4.1. In experiments in which same-donor samples were compared across two conditions, we performed a paired Student's *t*-test. Survival curves were compared using the log-rank Mantel–Cox test. Statistical methods were not used to predetermine sample sizes.

### Reporting summary

Further information on research design is available in the Nature Portfolio Reporting Summary linked to this article.

## Data availability

Transcription factor constructs will be made available through material transfer agreements when possible. The bulk RNA-seq, ATAC-seq and single-cell RNA-seq datasets were aligned to human genome hg38; they have been deposited in the NCBI Gene Expression Omnibus and are accessible through the accession number GSE255416. Source data are provided with this paper.

## Code availability

All code associated with this paper have been deposited to the Weber Lab GitHub repository (https://github.com/Weber-Lab-CHOP/FOXO1_2024)[59].

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

**Acknowledgements** We thank W. Yu for guidance with single-cell ATAC-seq data analysis; R. Majzner and F. Staback Rodriguez for thoughtful discussions; the National Cancer Institute at Frederick for providing the 1A7 anti-14g2a idiotype antibody; B. Jena and L. J. N. Cooper for providing the monoclonal anti-FMC63 idiotype antibody; and the High Throughput Sequencing Core at CHOP for help with sequencing. We also acknowledge the Flow Cytometry Core at CHOP and the Stanford Institute for Stem Cell Biology and Regenerative Medicine FACS core for equipment and technical support. Schematics were created with BioRender.com. This work was supported by the National Cancer Institute Immunotherapy Discover and Development (1U01CA232361-A1 to S.A.G. and E.W.W.; K08CA23188-01 and U01CA260852 to A.T.S.; and U54CA232568-01 to C.L.M.); the National Human Genome Research Institute (K99 HGHG012579 to C.A.L.); the Parker Institute for Cancer Immunotherapy (C.A.L., A.T.S., C.L.M. and E.W.W); the V Foundation for Cancer Research (E.W.W.); a Society for Immunotherapy of Cancer Rosenberg Scholar Award (E.W.W.); Stand Up 2 Cancer (St. Baldrick's) (NCI SU2CAACR-DT1113 to C.L.M.); the Virginia and D.K. Ludwig Fund for Cancer Research (C.L.M.); and NCI U2C CA233285 (K.T.). C.L.M., A.T.S., C.A.L. and E.W.W. are members of the Parker Institute for Cancer Immunotherapy, which supports cancer immunotherapy research at Stanford University and the University of Pennsylvania. Stand Up 2 Cancer is a program of the Entertainment Industry Foundation administered by the American Association for Cancer Research.

**Author contributions** E.W.W. and C.L.M. conceived the study, secured funding and supervised the project. A.E.D., K.P.M., G.T.R., B.D., J.L., Y.C., B.M., M.M., J.A.-U., R.H., A.W., P.X., D.K., G.Z., M.B., P.J.Q., Z.M., K.S., W.Z., F.R., M.L., J.H., S.E.M. and E.W.W. designed and performed wet-lab experiments. K.P.M., A.Y.C., B.D., A.W., G.M.C., and C.A.L. performed RNA-seq and ATAC-seq computational analyses. I.-Y.J. and J.A.F. performed analyses on and interpreted clinical data. K.T., S.A.G., J.A.F., E.S. and A.T.S. supervised experiments and analyses. A.E.D., K.P.M., E.S., C.L.M. and E.W.W. wrote the manuscript. All authors discussed the results and edited the manuscript.

**Competing interests** C.A.L. is a consultant to Cartography Biosciences. S.A.G. receives research funding from Novartis, Kite, Vertex and Servier; consults for Novartis, Roche, GSK, Humanigen, CBMG, Eureka, Janssen/JNJ and Jazz Pharmaceuticals; and has advised for Novartis, Adaptimmune, TCR2, Cellctis, Juno, Vertex, Allogene, Jazz Pharmaceuticals and Cabaletta. J.A.F. receives research funding from Tceleron (formerly Tmunity Therapeutics) and Danaher Corporation; consults for Retro Biosciences; and is a member of the scientific advisory boards of Cartography Biosciences and Shennon Biotechnologies. A.T.S. is a founder of Immunai and Cartography Biosciences and receives research funding from Allogene Therapeutics and Merck Research Laboratories. C.L.M. is a co-founder of and holds equity in Link Cell Therapies, Cargo Therapeutics (formerly Syncopation Life Sciences) and Lyell Immunopharma; holds equity and consults for Mammoth and Ensoma; consults for Immatics and Nektar; and receives research funding from Tune Therapeutics. E.W.W. holds equity in Lyell Immunopharma and consults for Umoja Immunopharma. The remaining authors declare no competing interests.

**Additional information**
**Correspondence and requests for materials** should be addressed to Crystal L. Mackall or Evan W. Weber.

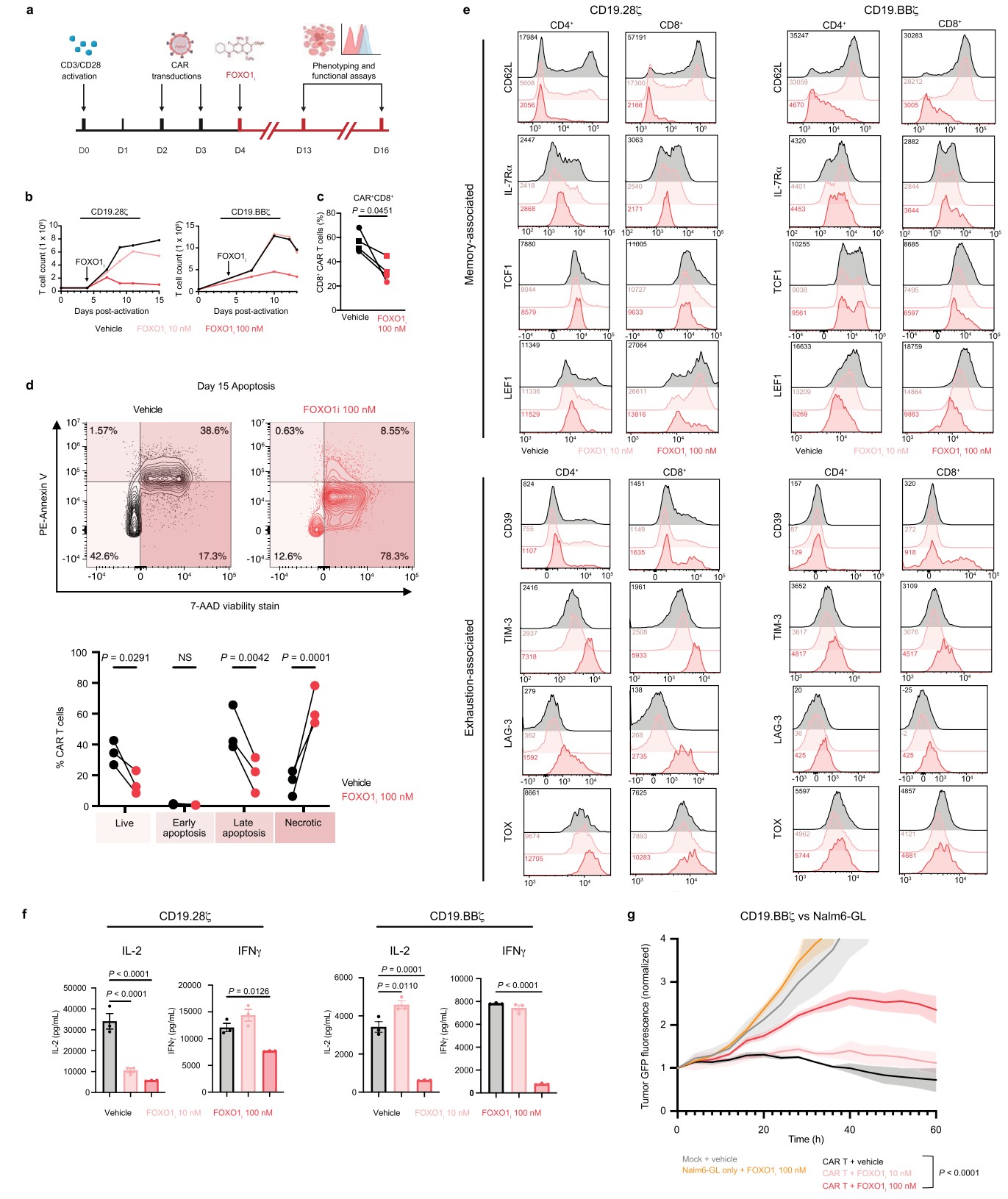

**Extended Data Fig. 1** | See next page for caption.

**Extended Data Fig. 1 | Pharmacological inhibition of FOXO1 impairs expansion, formation of a memory phenotype and antitumour function in CD19.28ζ and CD19.BBζ CAR T cells.** CAR T cells were treated with DMSO or 10 nM or 100 nM of the small molecule AS1842856 (FOXO1$_i$) starting on day 4 post-activation and treated every 2–3 days thereafter. **a**, Schematic of FOXO1$_i$ experimental model. **b**, CD19.28ζ (left) or CD19.BBζ (right) CAR T cell expansion ($n$ = 2 donors). **c**, Percent CD8$^+$ in CD19.28ζ (circles) and CD19.BBζ (squares) cells ($n$ = 2 donors for each CAR). **d**, Apoptosis in CD19.BBζ CAR T cells at day 15 post-activation. Contour plots show 1 representative donor and bar graphs show mean±s.e.m. of $n$ = 3 donors. **e**, Expression of memory- and exhaustion-associated markers on CD19.28ζ and CD19.BBζ cells. Histograms show 1 representative donor ($n$ = 2 donors). **f**, Cytokine secretion from CD19.28ζ and CD19.BBζ cells in response to Nalm6 cells. Graphs show mean±s.d. of triplicate wells from 1 representative donor ($n$ = 2 donors). **g**, Cytotoxicity of CD19.BBζ cells against Nalm6 cells at a 1:1 E:T ratio. Data is normalized to $t$ = 0 and show mean±s.d. of triplicate wells from 1 representative donor ($n$ = 2 donors). Statistics are shown for $t$ = 60 h. Statistical comparisons were performed using paired two-tailed Student's $t$-test (**c**), two-way ANOVA with Šídák's test (**d**) and one-way (**f**) or two-way (**g**) ANOVA with Dunnett's test. E:T ratio, effector:target cell ratio. NS, not significant.

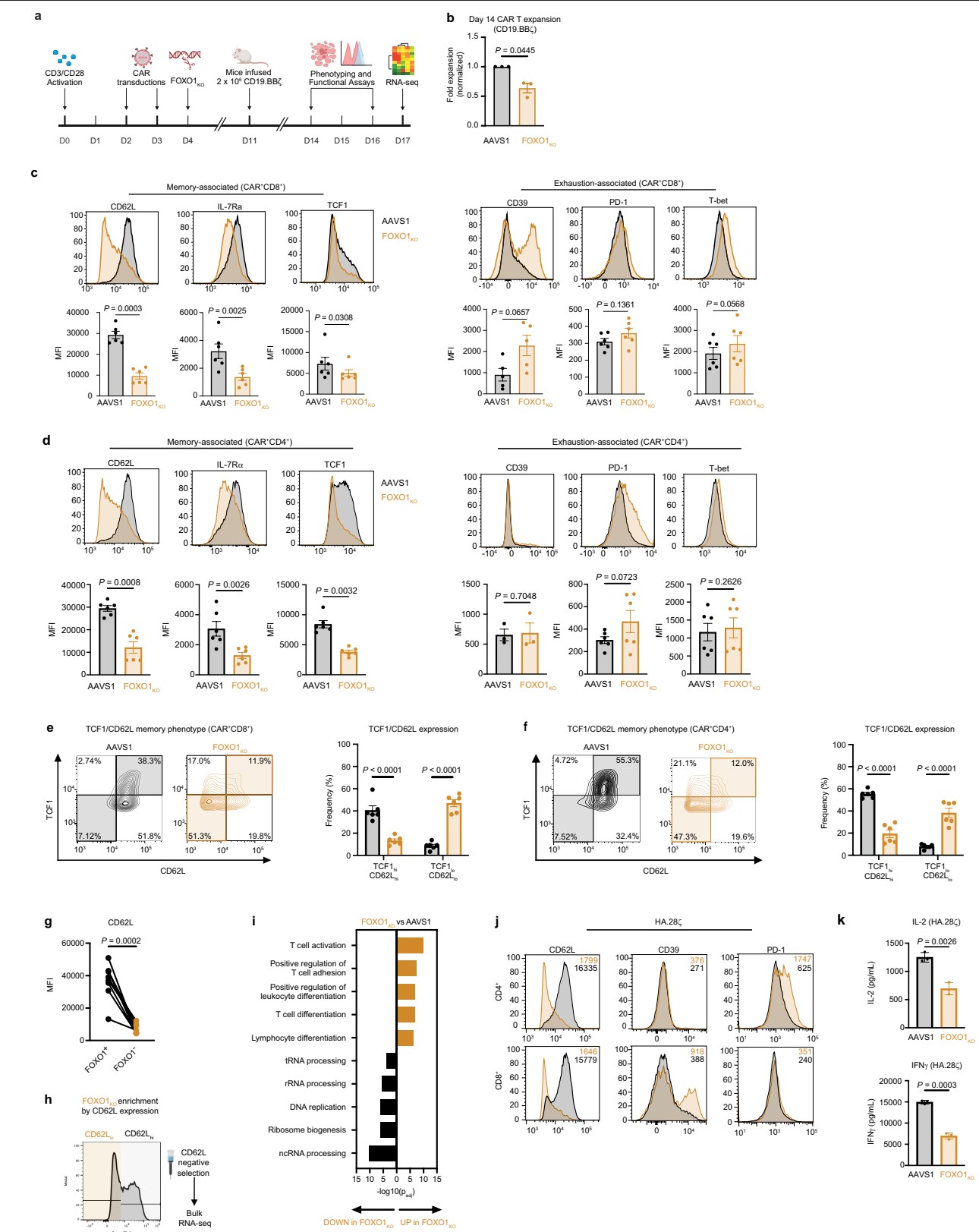

**Extended Data Fig. 2** | See next page for caption.

**Extended Data Fig. 2 | CRISPR knockout of FOXO1 attenuates memory formation and promotes exhaustion in CD19.BBζ and HA.28ζ CAR T cells.** **a**–**i**, CRISPR–Cas9 gene editing of *AAVS1* (AAVS1) or *FOXO1* (FOXO1$_{KO}$) in CD19. BBζ CAR T cells. **a**, Schematic depicting generation of FOXO1$_{KO}$ CAR T cells and downstream assays. **b**, Day 14 FOXO1$_{KO}$ expansion normalized to AAVS1. Data show mean±s.e.m. of $n$ = 3 donors. **c**–**f**, Flow cytometric analysis of memory- and exhaustion-associated markers on CD8$^+$ (**c**,**e**) and CD4$^+$ (**d**,**f**) CD19.BBζ cells. Histograms and contour plots show a representative donor and bar graphs show mean±s.e.m. of $n$ = 3-6 donors. CD62L, IL-7Rα, TCF1, and CD39 histograms in **c** also appear in Fig. 1c. **g**, MFI of CD62L in FOXO1$^+$ and FOXO1$^-$ gated subpopulations of CD19.BBζ cells. **h**, Schematic showing CD62L$_{lo}$/FOXO1$_{KO}$ cell

negative selection strategy for RNA-seq experiments. **i**, GO term analyses showing curated lists of up- and downregulated processes in FOXO1$_{KO}$ compared to AAVS1. Data show Benjamini–Hochberg-adjusted $P$ value ($n$ = 3 donors). **j**, Flow cytometric analysis of memory- and exhaustion-associated markers in day 15 HA.28ζ CAR T cells. Background-subtracted MFI is displayed. **k**, Cytokine secretion from day 15 HA.28ζ cells in response to Nalm6. Graphs show mean±s.d. of 3 technical replicates from one representative donor ($n$ = 2 donors). Statistical comparisons were performed using paired two-tailed Student's *t*-test (**b**,**c**,**d**,**g**), two-way ANOVA with Bonferroni's test (**e**,**f**), two-tailed Student's *t*-test (**k**) and one-sided hypergeometric test (**i**).

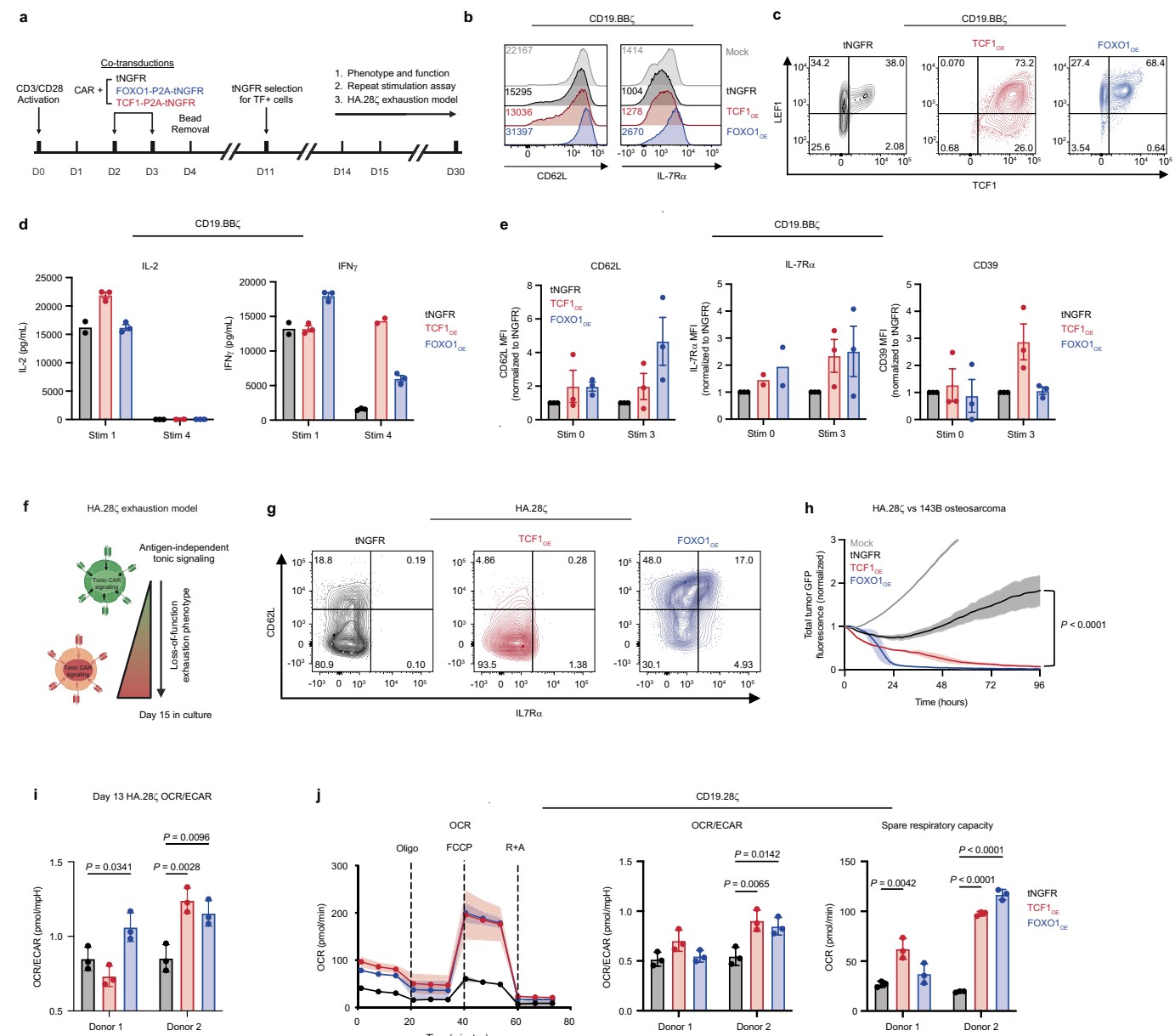

**Extended Data Fig. 3 | FOXO1 overexpression promotes a memory phenotype and mitigates exhaustion in CAR T cells. a**, Schematic depicting engineering of truncated NGFR-only (tNGFR), TCF1/tNGFR- (TCF1$_{OE}$), and FOXO1/tNGFR- (FOXO1$_{OE}$) CAR T cells and magnetic isolation of tNGFR$^+$ cells for downstream analyses. **b–e**, Phenotypic and functional analyses of CD19.BBζ CAR T cells at baseline and during repeat stimulation with Nalm6 cells. **b,c**, Flow cytometric analysis of CD62L and IL-7Rα (**b**) and TCF1 and LEF1 (**c**) from 1 representative donor (n = 4 donors). **d**, Cytokine secretion from CD19.BBζ cells after 1 or 4 stimulations with Nalm6 cells. Data show mean ± s.d. of 2–3 triplicate wells from 1 representative donor (n = 2 donors). **e**, Flow cytometric analysis of CD62L, IL-7Rα, and CD39 on tNGFR$^+$ CD8$^+$ CAR T cells prior to the first stimulation (Stim 0) and 7 days after the third stimulation (Stim 3). Data show mean ± s.e.m. of mean fluorescence intensity normalized to tNGFR levels from n = 2–3 donors. **f–i**, CAR T cell exhaustion model[7,23] whereby T cells express a high-affinity

GD2-targeting CAR (HA.28ζ) that promotes antigen-independent tonic CAR signalling. **f**, Model schematic. **g**, Flow cytometric analysis of day 15 CD62L and IL-7Rα. Data show 1 representative donor (n = 5 donors). **h**, Cytotoxicity of day 15 HA.28ζ cells against 143B cells at a 1:8 E:T ratio. Data is normalized to t = 0 and show mean ± s.d. of 3 triplicate wells from 1 representative donor (n = 3 donors). Statistics were performed at t = 96 h. **i,j**, Seahorse metabolic analyses on day 13 of culture (n = 2 donors). **i**, Ratio of OCR to ECAR of HA.28ζ cells. **j**, CD19.28ζ cell OCR (left), OCR to ECAR ratio (centre), and spare respiratory capacity (right). OCR line graph shows 1 representative donor. Bar graphs show mean ± s.d. of three representative time points within each donor. Statistical comparisons were performed using one-way ANOVA with Tukey's test (**h**) or Dunnett's test (**i,j**). E:T ratio, effector:target cell ratio. OCR, Oxygen consumption rate. ECAR, extracellular acidification rate.

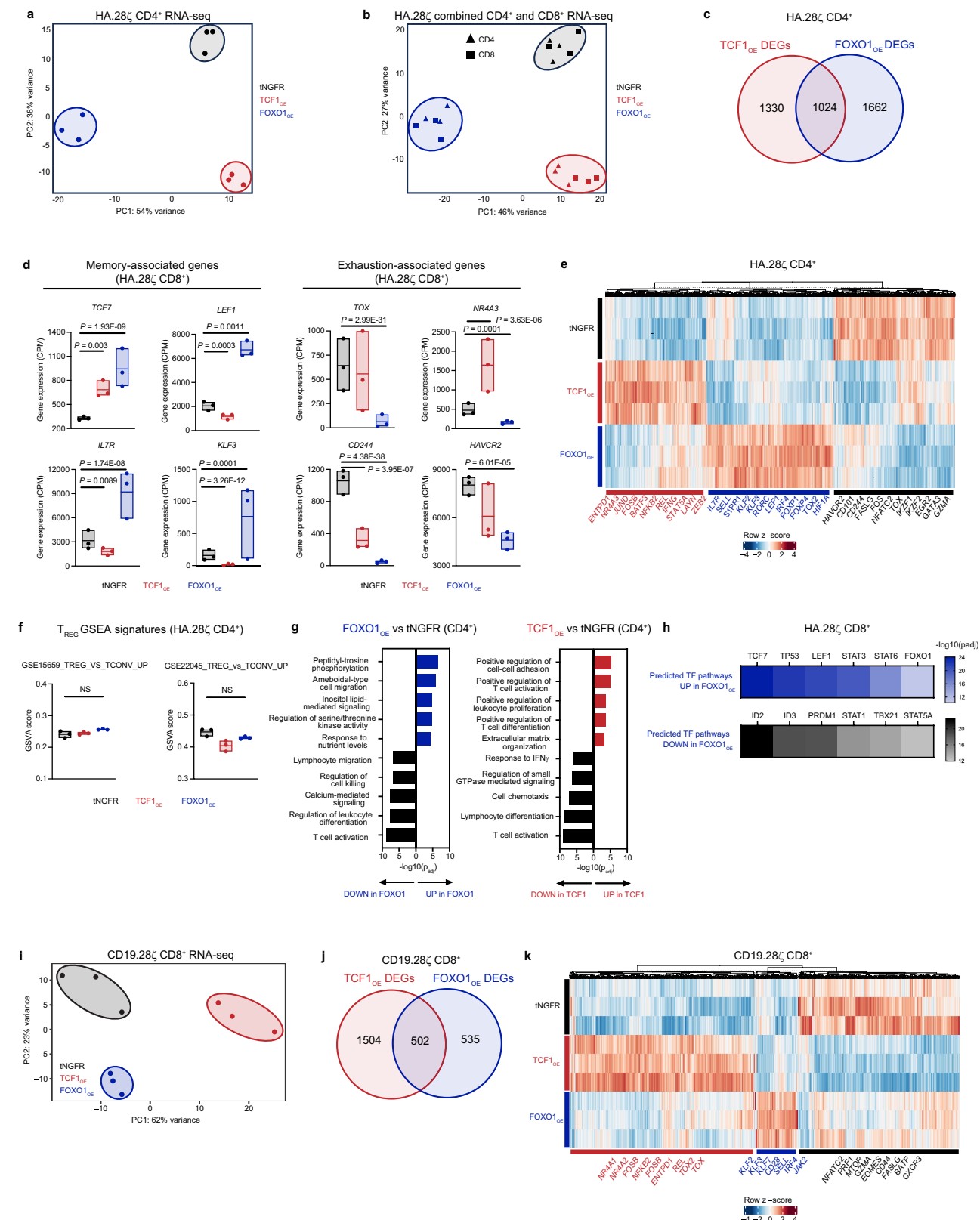

**Extended Data Fig. 4** | See next page for caption.

**Extended Data Fig. 4 | Overexpression of FOXO1 induces a memory-like transcriptional program in CAR T cells. a–g**, Bulk RNA-seq analyses of day 15 tNGFR$^+$ CD4$^+$ HA.28ζ CAR T cells overexpressing tNGFR, TCF1$_{OE}$, or FOXO1$_{OE}$ ($n$ = 3 donors). **a**, PCA of CD4$^+$ cells. **b**, PCA that includes CD4$^+$ samples plotted in **a** and CD8$^+$ samples plotted in Fig. 2a. **c**, Venn diagram showing the number of unique and shared DEGs in CD4$^+$ TCF1$_{OE}$ and FOXO1$_{OE}$ cells compared to tNGFR cells (Bonferroni-adjusted $P < 0.05$ with abs(log$_2$FC)>0.5). **d**, Expression of memory- and exhaustion-associated genes. Centre line represents the mean counts per million. **e**, Heat map and hierarchical clustering of DEGs. Genes of interest are shown. Scale shows normalized z-scores for each DEG. **f**, GSVA using published human CD4$^+$ regulatory T cell (T$_{reg}$) signatures[42,43]. Centre line represents mean score. **g**, GO term analyses showing curated lists of top up- and downregulated processes in CD4$^+$ FOXO1$_{OE}$ and TCF1$_{OE}$ cells versus tNGFR cells. Data show Benjamini–Hochberg-adjusted $P$. **h**, QIAGEN IPA of upregulated and downregulated TF pathways in FOXO1$_{OE}$ cells versus tNGFR cells. Data show adjusted $P$. **i–k**, Bulk RNA-seq analyses of day 15 tNGFR$^+$ CD8$^+$ CD19.28ζ cells overexpressing tNGFR, TCF1$_{OE}$, or FOXO1$_{OE}$ ($n$ = 3 donors). **i**, PCA analysis. **j**, Venn diagram showing the number of unique and shared DEGs in TCF1$_{OE}$ and FOXO1$_{OE}$ cells compared to tNGFR cells (Bonferroni-adjusted $P < 0.05$ with log$_2$(fold change) < 0.5). **k**, Heat map and hierarchical clustering of DEGs. Genes of interest are shown. Scale shows normalized z-scores for each DEG. Statistical comparisons were performed using DESeq2 (**c**,**d**,**e**,**j**,**k**), repeated-measures one-way ANOVA with Tukey's test (**f**) and one-sided hypergeometric test (**g**). NS, not significant.

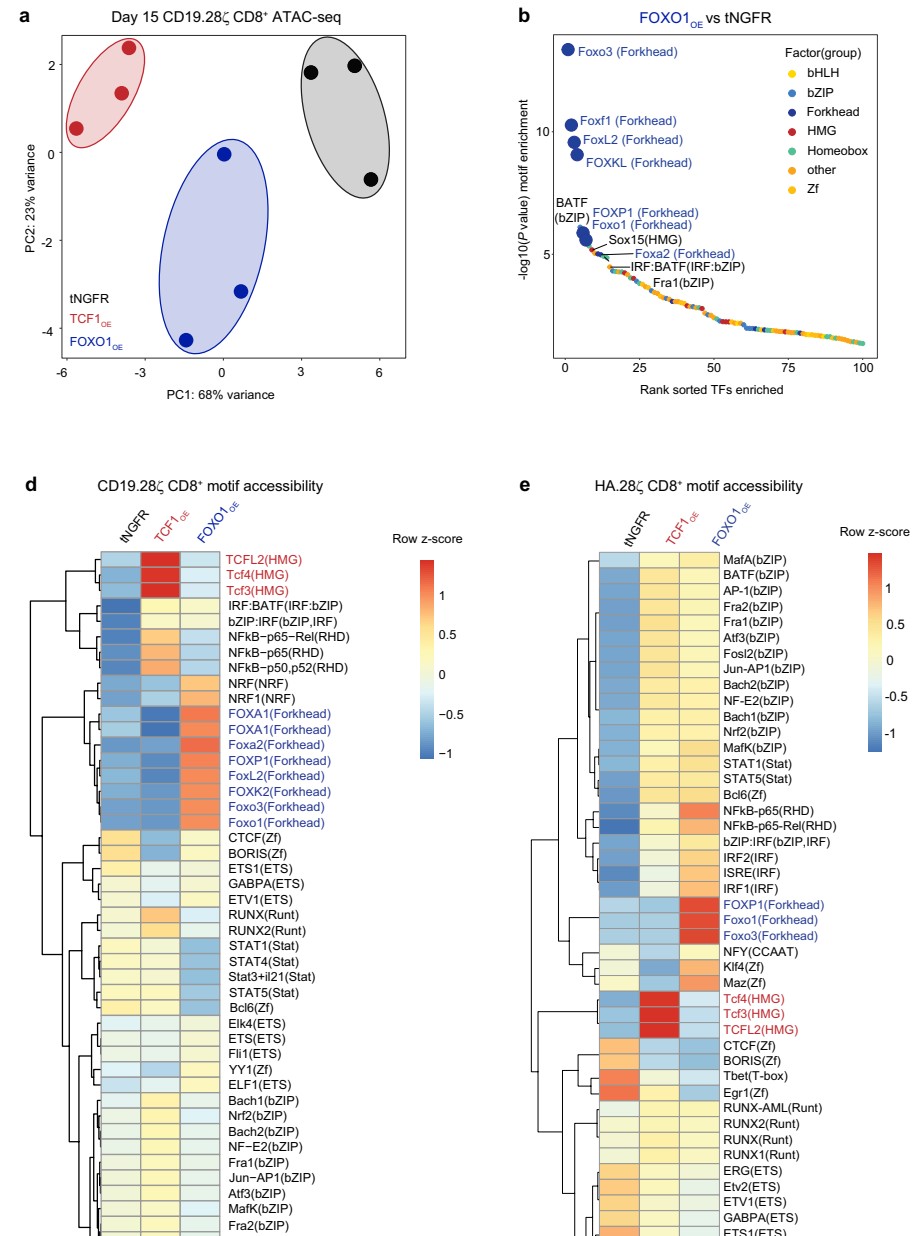

**Extended Data Fig. 5 | FOXO1 or TCF1 overexpression induces chromatin remodelling in CD19.28ζ and HA.28ζ CAR T cells. a–e**, Bulk ATAC-seq analyses of day 15 tNGFR⁺ CD8⁺ CAR T cells expressing either CD19.28ζ (**a–d**) or HA.28ζ (**e**) (*n* = 3 donors). **a**, PCA of CD19.28ζ cells. **b,c**, Rank-ordered plot of differentially accessible TF-binding motifs (*P* < 0.05 with abs(log₂FC)>0.5) in FOXO1_OE cells (**b**) and TCF1_OE cells (**c**) versus tNGFR cells. **d,e**, Heat maps and hierarchical clustering of mean differential motif accessibility of CD19.28ζ (**d**) or HA.28ζ (**e**) cells. Scales show normalized z-scores for each motif. Statistical comparisons were performed using DESeq2 (**b–e**).

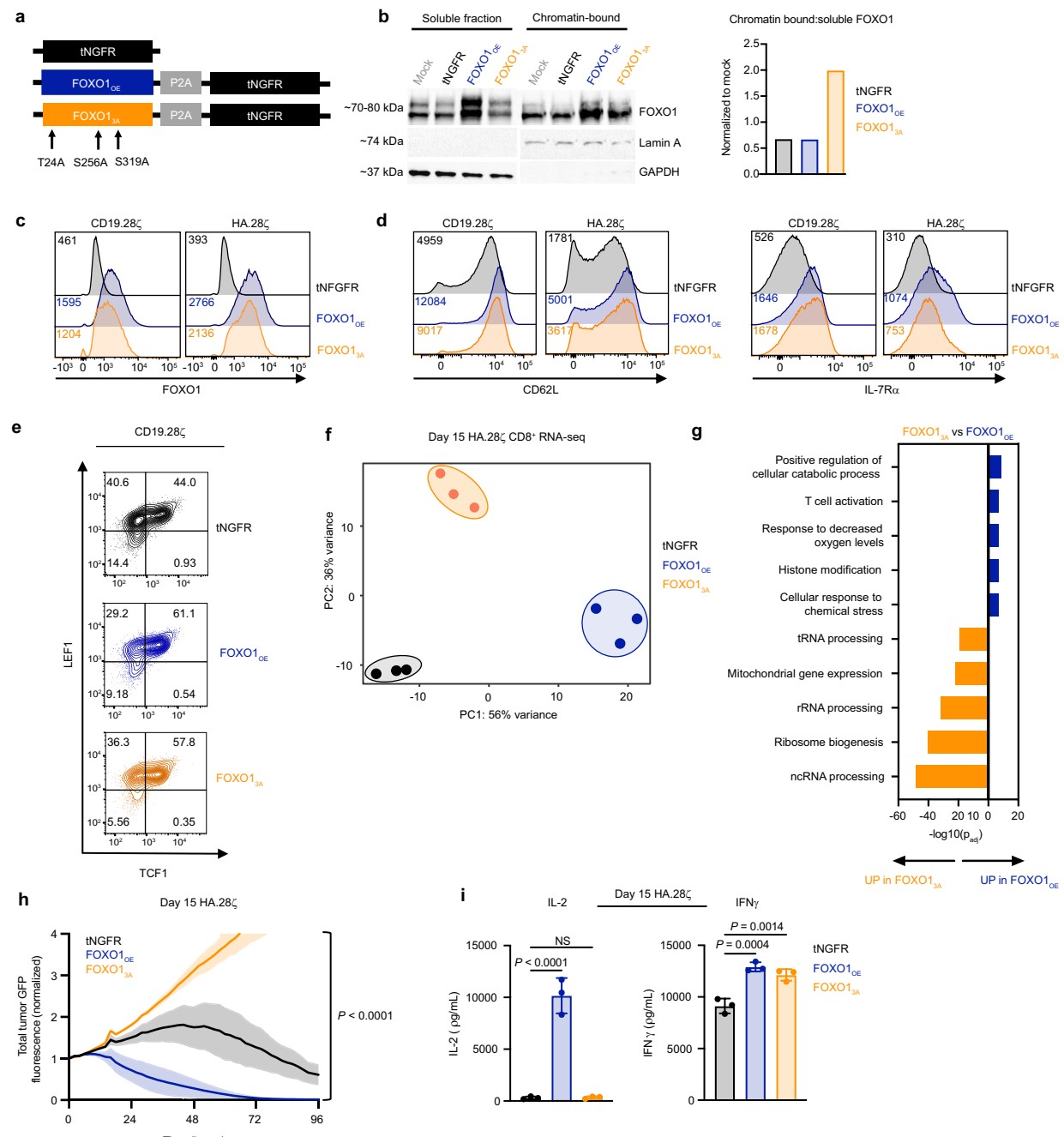

**Extended Data Fig. 6 | Nuclear-restricted FOXO1 promotes a memory-like phenotype but impairs effector function. a**, Schematic showing a mutated variant of FOXO1 that contains three amino acid substitutions (T24A, S256A, and S319A) which restrict nuclear export (FOXO1$_{3A}$). **b**, Analysis of soluble and chromatin-bound FOXO1 fractions isolated from tNGFR$^+$ non-CAR T cells that were activated with Dynabeads for 24 h prior to cell collection. Western blots (left) and bar graph (right) representing the ratio of chromatin-bound to soluble FOXO1 normalized to mock T cells are shown for 1 representative donor ($n = 2$ donors). **c**, FOXO1 expression in CD19.28ζ and HA.28ζ CAR T cells from 1 representative donor ($n = 5$ donors). **d**, CD62L and IL-7Rα expression in CD19.28ζ and HA.28ζ CAR T cells from 1 representative donor ($n = 3$ donors). **e**, TCF1 and LEF1 expression in CD19.28ζ CAR T cells from 1 representative

donor ($n = 3$ donors). **f,g**, RNA-seq on HA.28ζ CAR T cells. tNGFR and FOXO1$_{OE}$ samples are also represented in Fig. 2 and Extended Data Fig. 4. **f**, PCA. **g**, GO term analyses showing curated lists of top up- and downregulated processes in FOXO1$_{3A}$ vs FOXO1$_{OE}$. Data show Benjamini–Hochberg-adjusted $P$ value. **h**, Cytotoxicity of HA.28ζ cells against Nalm6 at a 1:1 E:T ratio. Data are normalized to $t = 0$ and show mean±s.d. from 1 representative donor ($n = 3$ donors). Statistics were performed at $t = 96$ h. **i**, Cytokine secretion from day 15 HA.28ζ CAR T cells in response to 143B cells. Plots show mean±s.d. of 3 wells from 1 representative donor ($n = 3$ donors). Statistical comparisons were performed using one-sided hypergeometric test (**g**), one-way ANOVA with Tukey's test (**h**) or Dunnett's test (**i**). E:T ratio, effector:target cell ratio.

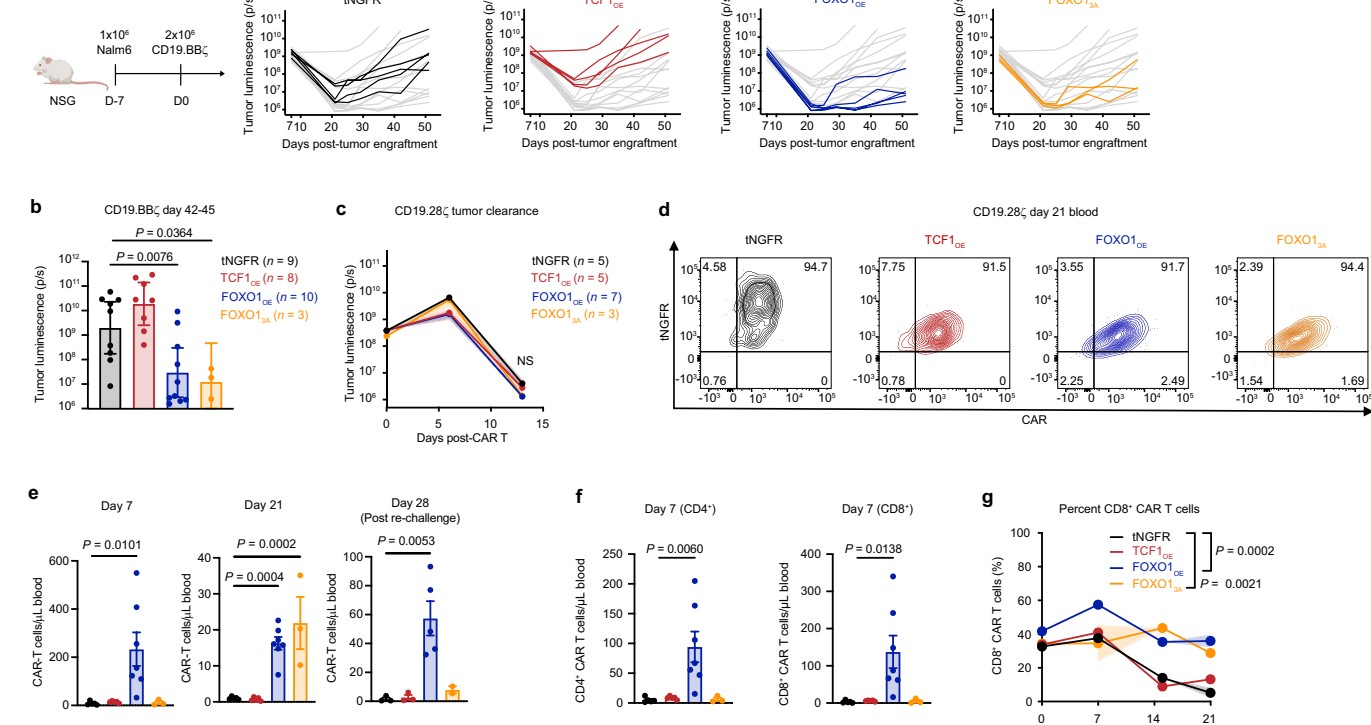

**Extended Data Fig. 7 | FOXO1$_{OE}$ CAR T cells show enhanced antitumour activity in leukaemia xenograft models. a,b,** A curative dose of 2×10[6] tNGFR[+] CD19.BBζ CAR T cells overexpressing tNGFR, TCF1$_{OE}$, FOXO1$_{OE}$, or FOXO1$_{3A}$ were infused into Nalm6 leukaemia-bearing mice 7 days post-engraftment (*n* = 2 donors tested in 2 independent experiments) **a**, Experimental schematic (left) and tumour bioluminescence of multiple time points (right) from 1 representative donor (*n* = 3-5 mice per group). **b**, Tumour bioluminescence from day 42-45. Data show mean±s.e.m. from 2 donors tested in 2 independent experiments (*n* = 3-10 mice per group; *n* = 1 donor for FOXO1$_{3A}$). **c–g**, A curative dose of 1×10[6] tNGFR[+] CD19.28ζ cells were infused into Nalm6-bearing mice 7 days post-engraftment. Mice were rechallenged with 10×10[6] CD19[+] or CD19[−] Nalm6 on day 21 post-CAR T cell infusion (*n* = 2 donors tested in 2 independent

experiments). **c**, Tumour bioluminescence over time. Data show mean±s.e.m. of *n* = 3–7 mice per group from 1 representative donor. **d**, CD19.28ζ and tNGFR expression on circulating CD45[+] CAR T cells on day 21. Contour plots show 1 representative mouse from each condition from 1 representative donor. **e**, Quantification of circulating CD45[+] CAR T cells on days 7, 21, and 28. **f**, CD4[+] and CD8[+] CAR T cells on day 7 (data derived from **e**). **g**, Percent CD8[+] CAR T cells. Graphs in **e–g** show mean±s.e.m. of *n* = 3–7 mice per group from 1 representative donor. Statistical comparisons were performed using nonparametric two-tailed Mann–Whitney test (**b**) and two-way (**c**) and one-way ANOVA with Dunnett's test (**e,f**) and mixed-effects model with Dunnett's test (**g**). NS, not significant.

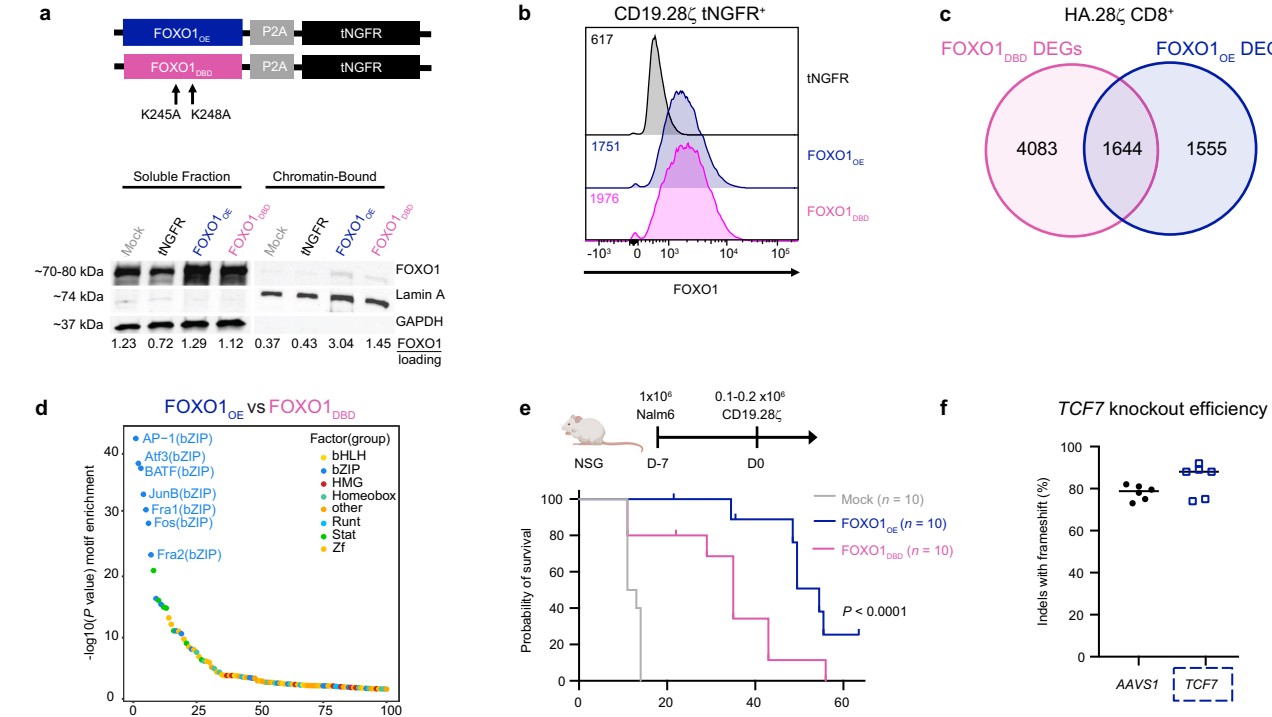

**Extended Data Fig. 8 | FOXO1$_{OE}$ reprogramming and enhanced antitumour activity are dependent on DNA binding. a**, Schematic depicting construct design and amino acid substitutions (K245A and K248A) to generate human FOXO1$_{DBD}$ (top) and western blots of indicated proteins in soluble and chromatin-bound fractions isolated from day 8 tNGFR$^+$ CD19.28ζ CAR T cells (bottom). Densitometry analyses are displayed below the blots. 1 representative donor from $n$ = 2 donors. **b**, FOXO1 expression in CD19.28ζ CAR T cells from one representative donor ($n$ = 5 donors). **c**, Bulk RNA-seq analyses of day 15 tNGFR$^+$ CD8$^+$ HA.28ζ CAR T cells show unique and shared DEGs in FOXO1$_{DBD}$ and FOXO1$_{OE}$ compared with tNGFR (Bonferroni-adjusted $P$ < 0.05 with abs(log$_2$FC)>0.5). FOXO1$_{OE}$ samples are also represented in Fig. 2 and Extended

Data Figs. 4 and 6. **d**, Bulk ATAC-seq of day 15 tNGFR$^+$ CD8$^+$ HA.28ζ CAR T cells. Rank-ordered plot of differentially accessible TF-binding motifs in FOXO1$_{OE}$ cells versus FOXO1$_{DBD}$ cells ($P$ < 0.05 with abs(log$_2$FC)>0.5). FOXO1$_{OE}$ samples are also represented in Fig. 2 and Extended Data Fig. 5. **e**, Schematic of stress test model (left) whereby Nalm6-engrafted mice were treated with mock T cells or FOXO1$_{OE}$ or FOXO1$_{DBD}$ CD19.28ζ CAR T cells. Survival curve shows pooled data from 2 donors tested in 2 independent experiments ($n$ = 10 mice per group, FOXO1$_{OE}$ data from 1 donor are also represented in Fig. 3a). **f**, *TCF7* knockout efficiency for bulk RNA-seq data corresponding to Fig. 3f,g. Data show the mean of $n$ = 3 donors with 2 technical replicates per donor. Statistical comparisons were performed using DESeq2 (**c**,**d**) and log-rank Mantel–Cox test (**e**).

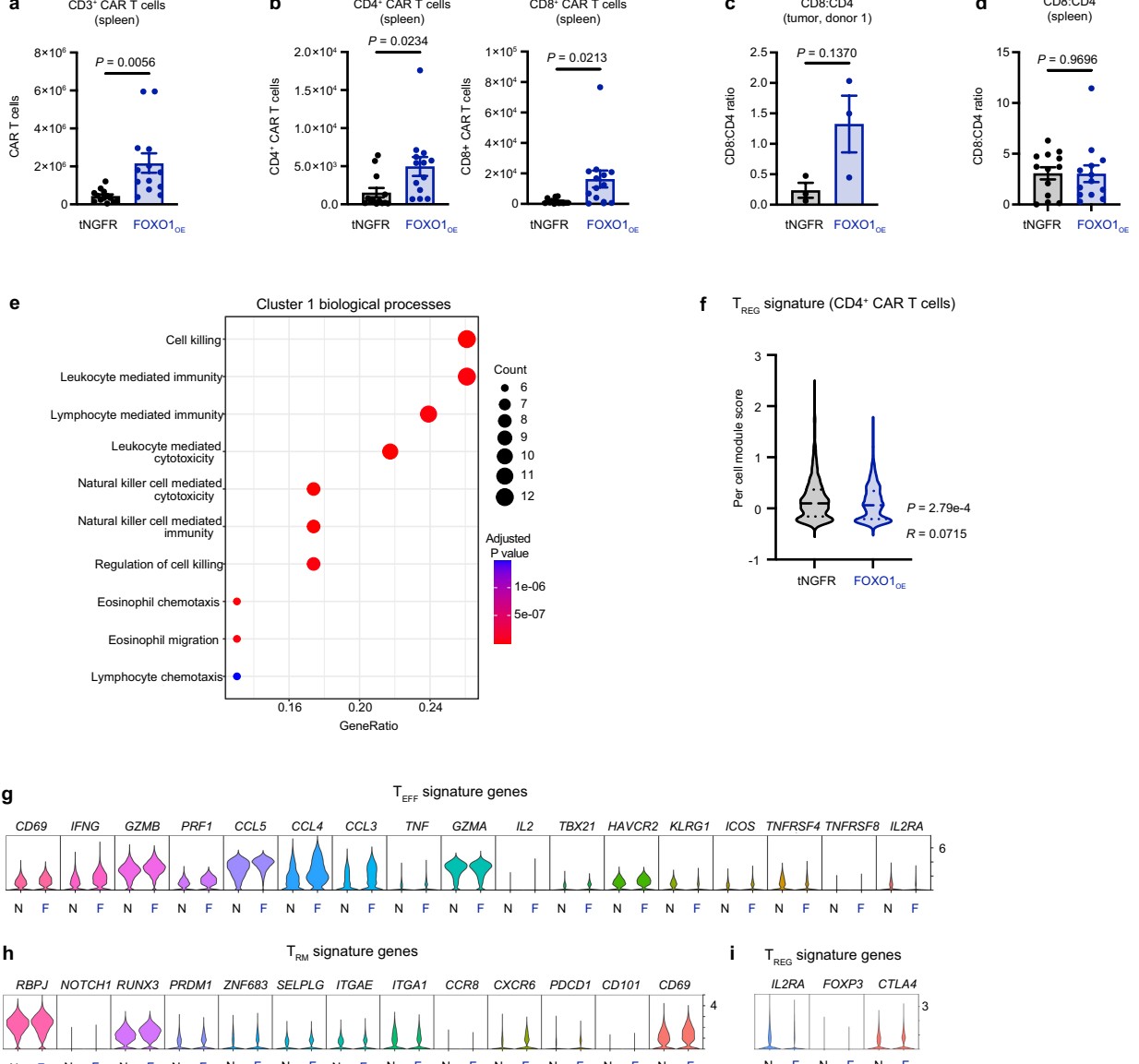

**Extended Data Fig. 9 | FOXO1$_{OE}$ CAR T cells exhibit improved persistence and effector- and tissue-residence-associated transcriptomic signatures in a solid-tumour xenograft model.** $5×10^6$ Her2.BBζ CAR T cells were infused into 143B-bearing mice 3 days post-engraftment. Tumours and spleens were collected on day 29 post-engraftment for phenotypic, functional, and sequencing-based assays. **a**, Total splenic CAR T cells. **b**, Total CD4$^+$ (left) and CD8$^+$ (right) splenic CAR T cells. **c**, Ratio of CD8$^+$ to CD4$^+$ tumour-infiltrating CAR T cells from donor 1 ($n$ = 3 mice per group). Donor 2 is shown in Fig. 4d. **d**, Ratio of CD8$^+$ to CD4$^+$ CAR T cells from spleens. Data in **a–d** show mean±s.d. of $n$ = 13 mice per group from 2 donors tested in 2 independent experiments

unless otherwise stated. **e–i**, Single-cell RNA-seq on day 29 tumour-infiltrating FOXO1$_{OE}$ or tNGFR cells. Cells were sorted and pooled from $n$ = 5 mice per group from 1 donor. **e**, Top enriched GO terms in Cluster 1, which was biased towards FOXO1$_{OE}$ cells. Gene ratio and Benjamini–Hochberg-adjusted $P$ value are shown. **f**, T$_{reg}$ transcriptional signature[58] score. **g**, T$_{eff}$ signature genes corresponding to T$_{eff}$ scores in Fig. 4i. **h**, T$_{RM}$ signature genes corresponding to T$_{RM}$ scores in Fig. 4i. **i**, T$_{reg}$ signature genes corresponding to scores in **f**. Statistical comparisons were performed using two-tailed Student's $t$-test (**a-d**), one-sided hypergeometric test (**e**) and two-sided Wilcoxon rank-sum test (**f**).

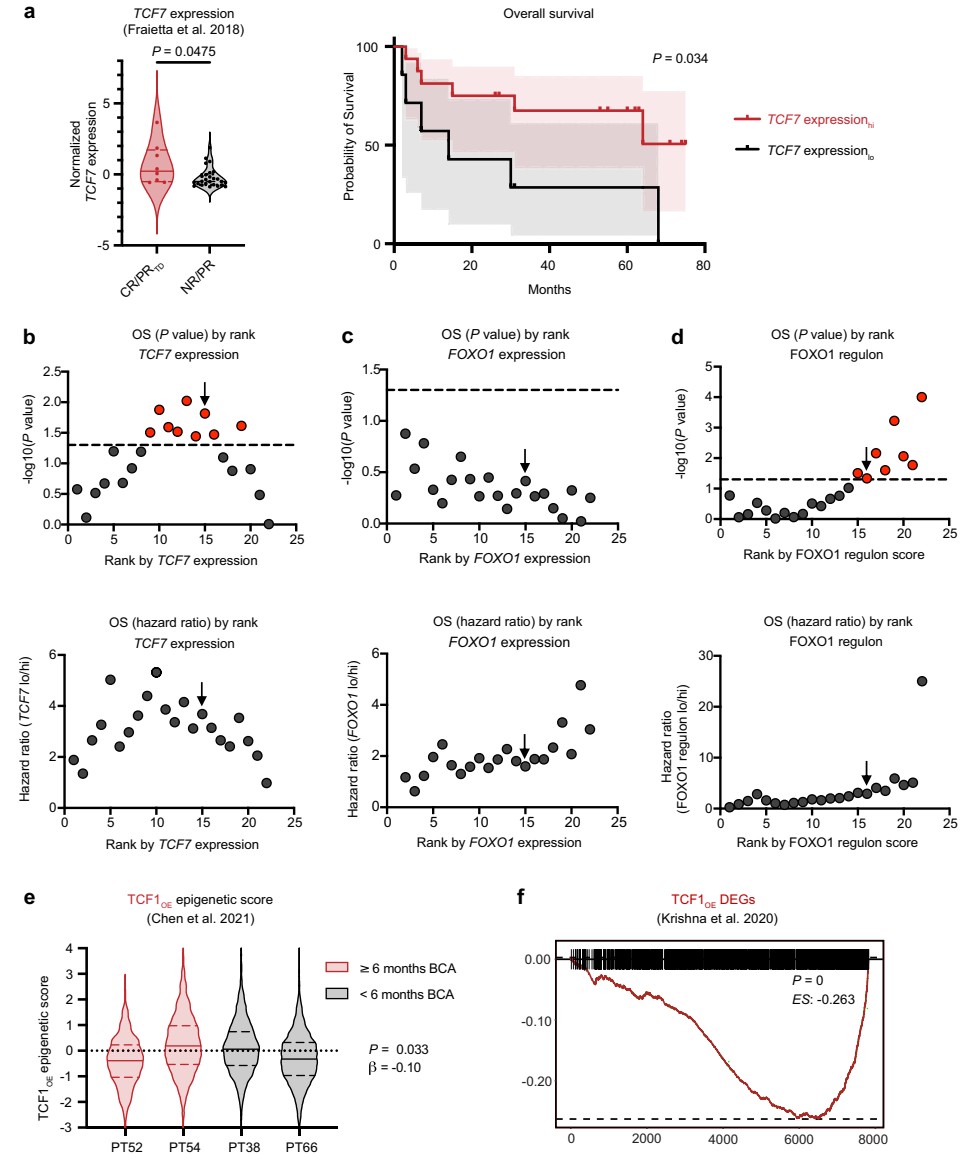

**Extended Data Fig. 10 | Endogenous *TCF7* transcript and FOXO1 regulon, but not TCF$_{OE}$ transcriptional or epigenetic signatures, predict CAR T cell and TIL responses in patients. a**, ssGSEA on RNA-seq from CAR-stimulated CTL019 cells[2] (complete responder, CR, partial responder with transformed disease, PR$_{TD}$, $n$ = 3; partial responder, PR, $n$ = 5; non-responder, NR, $n$ = 21). Enrichment score stratification points for patient survival analyses were determined using previously published methods[60]. **a**, *TCF7* expression is shown for patient outcomes (left) and overall survival (right). **b**–**d**, *P* values (top) and hazard ratios (bottom) of different stratification points in relation to overall survival (OS) of *TCF7* expression (**b**), *FOXO1* expression (**c**) and FOXO1 regulon (**d**). Dotted lines are drawn at *P* < 0.05 and black arrows indicate the stratification

points used. **e**, An epigenetic signatured derived from CD8$^+$ CD19.28ζ FOXO1$_{OE}$ bulk ATAC-seq data was applied to pre-manufactured paediatric CAR T cell single-cell ATAC-seq data[5]. Violin plots show TCF1$_{OE}$ epigenetic signature scores for patients with durable (Patient 52, $n$ = 616 cells; Patient 54, $n$ = 2959 cells) and short (Patient 38, $n$ = 2093 cells; Patient 66, $n$ = 2355 cells) CAR T cell persistence. **f**, GSEA was performed with CD8$^+$ HA.28ζ TCF1$_{OE}$ DEGs and DEGs derived from CD39$^-$CD69$^-$ patient TILs in adult melanoma[8]. Violin plots in **a,e** show minima and maxima; centre lines represent mean; dashed lines represent top and bottom quartiles. Statistical comparisons were performed using two-tailed Mann–Whitney test (**a**, left), log-rank Mantel–Cox test (**a**, right), two-sided Wald test (**e**), and two-sided Kolmogorov–Smirnov test (**f**).

| | |
|---|---|

# Reporting Summary

## Statistics

For all statistical analyses, confirm that the following items are present in the figure legend, table legend, main text, or Methods section.

| n/a | Confirmed | |
|---|---|---|
| ☐ | ☒ | The exact sample size (*n*) for each experimental group/condition, given as a discrete number and unit of measurement |
| ☐ | ☒ | A statement on whether measurements were taken from distinct samples or whether the same sample was measured repeatedly |
| ☐ | ☒ | The statistical test(s) used AND whether they are one- or two-sided<br>*Only common tests should be described solely by name; describe more complex techniques in the Methods section.* |
| ☐ | ☒ | A description of all covariates tested |
| ☐ | ☒ | A description of any assumptions or corrections, such as tests of normality and adjustment for multiple comparisons |
| ☐ | ☒ | A full description of the statistical parameters including central tendency (e.g. means) or other basic estimates (e.g. regression coefficient) AND variation (e.g. standard deviation) or associated estimates of uncertainty (e.g. confidence intervals) |
| ☐ | ☒ | For null hypothesis testing, the test statistic (e.g. *F*, *t*, *r*) with confidence intervals, effect sizes, degrees of freedom and *P* value noted<br>*Give P values as exact values whenever suitable.* |
| ☒ | ☐ | For Bayesian analysis, information on the choice of priors and Markov chain Monte Carlo settings |
| ☐ | ☒ | For hierarchical and complex designs, identification of the appropriate level for tests and full reporting of outcomes |
| ☐ | ☒ | Estimates of effect sizes (e.g. Cohen's *d*, Pearson's *r*), indicating how they were calculated |

*Our web collection on statistics for biologists contains articles on many of the points above.*

## Software and code

Policy information about availability of computer code

| | |
|---|---|
| Data collection | Flow cytometry: Data was collected on a BD Fortessa running FACS Diva version 8.0.1 or a Cytek Aurora running SpectroFlo Software version 3.1.0.<br>Cytokine secretion: Data was collected on a Tecan Spark plate reader or a BioTek Synergy H1 running Gen5 version 2.00.18<br>Killing assays: Image collection and analysis was performed on EssenBioscience/Sartorius IncuCyte ZOOM S3 Software<br>Immunoblotting: Image collection was performed on a ChemiDoc using Image Lab Touch Software version 3.0.1.14<br>In vivo analysis: Images were collected and analyzed using Perkin Elmer Living Image version 4.7.3 or Spectral Instruments Imaging Aura version 4.0.7 |
| Data analysis | Figures and Statistical Analysis: Figures were created and statistical tests ere performed on Graphpad Prism version 9.3.1<br>DNA sequence analysis and cloning: DNA sequences were analyzed on SnapGene version 6.0.5<br>Flow cytometry: Flow cytometric data was analyzed  FlowJo version 10.8.1<br>Cytokine secretion and killing assays: Data was analyzed on Graphpad Prism version version 9.4.1<br>Seahorse: Data was analyzed on Agilent Seahorse Wave Desktop Software<br>Immunoblotting: Data were using FIJI version 2.14.0/1.5f.<br>RNA-seq: The following analyses were performed on R, version 4.1.0: RNA-seq analysis was performed as per the nf-core RNAseq pipeline version 3.1.1 (https://github.com/nf-core/rnaseq). GSVA scores were calculated via the GSVA pipeline version 1.46.0 (https://github.com/rcastelo/GSVA). Both RNA-seq and ATAC-seq samples were analyzed via DESeq2 version 3.16. Motif search was performed utilizing HOMER version 4.11. Pathway enrichment analyses were performed using clusterProfiler version 4.6.2. Ingenuity Pathway Analysis was performed using QIAGEN Ingenuity Pathway Analysis 2022 Winter Release.<br>ATAC-seq: ATAC-seq analysis was performed as per the PEPATAC pipeline (https://pepatac.databio.org/en/latest/). ATAC enrichment was |

performed using gchromVAR (https://github.com/caleblareau/gchromVAR).

Single-cell RNA-seq: FASTQ files were generated and aligned to the genome with Cellranger version 7.1.0. Low quality cells with <300 or >7500 genes or >10% mitochondrial reads were removed using Seurat version 4.3.0. Doublets were identified using DoubletFinder v2.0.3. FindAllMarkers (Seurat) was used to identify differentially expressed (DE) genes in each cluster, and gene ontology (GO) analyses were performed for each cluster using ClusterProfiler version 4.6.2.

FOXO1 Regulon: Regulon analyses on single-cell ATAC-seq data were performed using Seurat v 4.3.0 and GSVA v1.46.0 and gchromVAR.

Code availability statement: All code associated with this paper are deposited to the Weber Lab GitHub (https://github.com/Weber-Lab-CHOP/FOXO1_2024).

For manuscripts utilizing custom algorithms or software that are central to the research but not yet described in published literature, software must be made available to editors and reviewers. We strongly encourage code deposition in a community repository (e.g. GitHub). See the Nature Portfolio guidelines for submitting code & software for further information.

# Data

Policy information about availability of data

All manuscripts must include a data availability statement. This statement should provide the following information, where applicable:
- Accession codes, unique identifiers, or web links for publicly available datasets
- A description of any restrictions on data availability
- For clinical datasets or third party data, please ensure that the statement adheres to our policy

Data availability statement: Transcription factor constructs will be made available through Material Transfer Agreements when possible. The bulk RNA-seq, ATAC-seq, and single-cell RNA-seq datasets were aligned to human genome hg38 and have been deposited in the NCBI Gene Expression Omnibus (GEO) and are accessible through the accession number GSE255416.

# Research involving human participants, their data, or biological material

Policy information about studies with human participants or human data. See also policy information about sex, gender (identity/presentation), and sexual orientation and race, ethnicity and racism.

| | |
|---|---|
| Reporting on sex and gender | *Use the terms sex (biological attribute) and gender (shaped by social and cultural circumstances) carefully in order to avoid confusing both terms. Indicate if findings apply to only one sex or gender; describe whether sex and gender were considered in study design; whether sex and/or gender was determined based on self-reporting or assigned and methods used.*<br>*Provide in the source data disaggregated sex and gender data, where this information has been collected, and if consent has been obtained for sharing of individual-level data; provide overall numbers in this Reporting Summary. Please state if this information has not been collected.*<br>*Report sex- and gender-based analyses where performed, justify reasons for lack of sex- and gender-based analysis.* |
| Reporting on race, ethnicity, or other socially relevant groupings | *Please specify the socially constructed or socially relevant categorization variable(s) used in your manuscript and explain why they were used. Please note that such variables should not be used as proxies for other socially constructed/relevant variables (for example, race/ethnicity should not be used as a proxy for socioeconomic status).*<br>*Provide clear definitions of the relevant terms used, how they were provided (by the participants/respondents, the researchers, or third parties), and the method(s) used to classify people into the different categories (e.g. self-report, census or administrative data, social media data, etc.)*<br>*Please provide details about how you controlled for confounding variables in your analyses.* |
| Population characteristics | Healthy human donor primary T cells were obtained from the Human Immunology Core at the Perelman School of Medicine at the University of Pennsylvania or from the Stanford Blood Center. |
| Recruitment | *Describe how participants were recruited. Outline any potential self-selection bias or other biases that may be present and how these are likely to impact results.* |
| Ethics oversight | *Identify the organization(s) that approved the study protocol.* |

Note that full information on the approval of the study protocol must also be provided in the manuscript.

# Field-specific reporting

Please select the one below that is the best fit for your research. If you are not sure, read the appropriate sections before making your selection.

☒ Life sciences      ☐ Behavioural & social sciences      ☐ Ecological, evolutionary & environmental sciences

For a reference copy of the document with all sections, see nature.com/documents/nr-reporting-summary-flat.pdf

# Life sciences study design

All studies must disclose on these points even when the disclosure is negative.

| | |
|---|---|
| Sample size | No sample size calculations were performed; group sizes were validated by experience with well-established, previously published models (1,2).<br><br>1: Lynn, R. C., Weber, E. W., Sotillo, E., Gennert, D., Xu, P., Good, Z., ... & Mackall, C. L. (2019). c-Jun overexpression in CAR T cells induces exhaustion resistance. Nature, 576(7786), 293-300.<br>2: Weber, E. W., Parker, K. R., Sotillo, E., Lynn, R. C., Anbunathan, H., Lattin, J., ... & Mackall, C. L. (2021). Transient rest restores functionality in exhausted CAR-T cells through epigenetic remodeling. Science, 372(6537), eaba1786. |
| Data exclusions | In the experiment referred to in Figure 5B,C, D, and E, 2 mice in each of the CD19.28ζ tNGFR, CD19.28ζ FOXO1OE, CD19.28ζ FOXO13A, and CD19.28ζ TCF1OE conditions had to be euthanized and data from these mice were excluded due to a non-tumor-related infectious disease complication. |
| Replication | T cells derived from least 2 different healthy donors were used for each experiment and were tested in a minimum of 2 independent experiments. For experiments where one representative donor was shown, data were representative of all donors. All attempts at replication were successful with the exception of the in vivo experiments noted above in "Data Exclusions" and one CD19.BBζ repeat stimulation experiment as per Figure 2C-F due to extremely low starting numbers of CD8+ T cells in one specific donor that interfered with downstream assays. |
| Randomization | For in vivo experiments, mice were randomized prior to CAR T cell infusion to ensure equal tumor burden across groups. For other experiments that involved CAR T cell engineering, bulk CD3+ cells from each healthy donor were randomly distributed into wells prior to viral transduction to ensure equal cellular heterogeneity across groups. |
| Blinding | In vivo tumor engraftment and T cell infusion were performed by technicians who were blinded to treatments and expected outcomes. Full blinding was not performed for other experiments. Fully-blinded experiments were not possible due to a limited number of investigators capable of performing such experiments. |

# Reporting for specific materials, systems and methods

We require information from authors about some types of materials, experimental systems and methods used in many studies. Here, indicate whether each material, system or method listed is relevant to your study. If you are not sure if a list item applies to your research, read the appropriate section before selecting a response.

## Materials & experimental systems

| n/a | Involved in the study |
|---|---|
| ☐ | ☒ Antibodies |
| ☐ | ☒ Eukaryotic cell lines |
| ☒ | ☐ Palaeontology and archaeology |
| ☐ | ☒ Animals and other organisms |
| ☒ | ☐ Clinical data |
| ☒ | ☐ Dual use research of concern |
| ☒ | ☐ Plants |

## Methods

| n/a | Involved in the study |
|---|---|
| ☒ | ☐ ChIP-seq |
| ☐ | ☒ Flow cytometry |
| ☒ | ☐ MRI-based neuroimaging |

## Antibodies

| | |
|---|---|
| Antibodies used | For Flow Cytometry:<br>From BD:<br>Anti-CD4 BUV395 (clone: SK3, catalog: 563550); Anti-CD8 BUV805 (clone: SK1, catalog: 612889); Anti-Blimp1 PE-CF594 (clone: 6D3, catalog: 565274); Anti-CD271 BV711 (clone: C40-1457, catalog: 743360); Anti-CD271 BV421 (clone: C40-1457, catalog: 562562); Anti-CD45RA FITC (clone: HI100, catalog: 561882).<br><br>From BioLegend:<br>Anti-CD62L BV605 (clone: DREG-56, catalog: 304834); Anti-CD45RA AF488 (clone: HI100, catalog: 304114); Anti-CD45RA BV711 (clone: HI100, catalog: 304137); Anti-IL7Ra BV421 (clone: A019D5, catalog: 351310); Anti-CD39 BV711 (clone: A1, catalog: 328228); Anti-CD39 APC-Cy7 (clone: A1, catalog: 328225); Anti-TIM3 BV510 (clone: F38-2E2, catalog: 345030); Anti-Tbet BV711 (clone: 4B10, catalog: 644820); Anti-Tbet BV785 (clone: 4B10, catalog: 644835); Anti-CD127 BV711 (clone: A019D5, catalog: 351327); Anti-CD8 AF700 (clone: SK1, catalog: 344723); Anti-CD62L PerCP-Cy5.5 (clone: DREG-56, catalog: 304823)<br><br>From Cell Signaling:<br>Anti-FOXO1 AF488 (clone: C29H4, catalog: 58223S); Anti-TCF1 PE (clone: C63D9, catalog: 14456); Anti-TCF1 AF647 (clone: C63D9, |

catalog: 6709S); Anti-Bcl-2 PE (clone: 124, catalog: 26295S); Anti-LEF1 AF488 (clone: C12A5, catalog: 8490S); Anti-LEF1 PE (clone: C12A5, catalog: 14440)

From eBiosciences:
Anti-PD1 PE-Cy7 (clone: J105, catalog: 25-2799-42); Anti-LAG3 PE (clone: 3DS223H, catalog: 12-2239-42)

From Invitrogen:
Anti-CD45 PerCP-Cyanine5.5 (clone: HI30, catalog: 45-0459-42)

Custom antibodies:
Sourced from the National Cancer Institute: Anti-14G2a CAR (clone 1A7, conjugated to Dylight 650 using Thermo Scientific Dylight 650 Labeling Kit catalog #84535)
Sourced from Genscript via custom prep: Anti-CD19 CAR (clone FMC63, conjugated to Dylight 650 using Thermo Scientific Dylight 650 Labeling Kit catalog #84535)

For cell selection:
From BD:
Anti-CD62L PE (clone: DREG-56, catalog: 555544)

From Biolegend:
Anti-CD271 Biotin (clone: ME20.4, catalog: 345122)

For Western Blot:
From Cell Signaling:
Anti-FOXO1 (clone: C29H4, catalog: 2880), Anti-Lamin A (clone 133A2, catalog: 86846), Anti-GAPDH (clone: D4C6R, catalog: 97166), Anti-rabbit IgG, HRP-linked Antibody (clone: n/a, catalog: 7074), Anti-mouse IgG, HRP-linked Antibody (clone: n/a, catalog: 7076)

Validation

All flow cytometry antibodies were validated by manufacturers on various human peripheral blood mononuclear cells except anti-CD271 antibodies which were validated on human neuroblastoma cell line SK-N-MC, which express a high level of NGFR.
Antibodieswere additionally validated at Stanford or Children's Hospital of Philadelphia by comparing antibody-specific staining to isotype and unstained controls.
Western blot antibodies were validated by manufacturers on cell lines as noted below in the specific antibody sections.

Antibody validation can be found at the following sites:

Anti-CD4-BUV395: https://www.bdbiosciences.com/en-us/products/reagents/flow-cytometry-reagents/research-reagents/single-color-antibodies-ruo/buv395-mouse-anti-human-cd4.563552
Anti-CD8-BUV805: https://www.bdbiosciences.com/en-us/products/reagents/flow-cytometry-reagents/research-reagents/single-color-antibodies-ruo/buv805-mouse-anti-human-cd8.612889
Anti-CD62L-BV605:
https://www.bdbiosciences.com/en-us/products/reagents/flow-cytometry-reagents/research-reagents/single-color-antibodies-ruo/bv605-mouse-anti-human-cd62l.562719
Anti-Blimp-1-PE-CF594:
https://www.bdbiosciences.com/en-us/products/reagents/flow-cytometry-reagents/research-reagents/single-color-antibodies-ruo/pe-cf594-rat-anti-blimp-1.565274
Anti-CD271-BV711: https://www.bdbiosciences.com/en-us/products/reagents/flow-cytometry-reagents/research-reagents/single-color-antibodies-ruo/bv711-mouse-anti-human-cd271.743360
Anti-CD271-BV421: https://www.bdbiosciences.com/en-us/products/reagents/flow-cytometry-reagents/research-reagents/single-color-antibodies-ruo/bv421-mouse-anti-human-cd271.562562
Anti-CD45RA-FITC:
https://www.bdbiosciences.com/en-us/products/reagents/flow-cytometry-reagents/research-reagents/single-color-antibodies-ruo/fitc-mouse-anti-human-cd45ra.561882
Anti-CD62L-BV605:
https://www.biolegend.com/en-us/products/brilliant-violet-605-anti-human-cd62l-antibody-8554?GroupID=BLG10034
Anti-CD45RA-AF488:
https://www.biolegend.com/en-us/products/alexa-fluor-488-anti-human-cd45ra-antibody-3337?GroupID=GROUP658
Anti-CD45RA-BV711:
https://www.biolegend.com/en-us/products/brilliant-violet-711-anti-human-cd45ra-antibody-7937
Anti-IL-7Ra-BV421:
https://www.biolegend.com/en-us/products/brilliant-violet-421-anti-human-cd127-il-7ralpha-antibody-7155
Anti-CD39-BV711:
https://www.biolegend.com/en-us/products/brilliant-violet-711-anti-human-cd39-antibody-1390
Anti-CD39-APC-Cy7:
https://www.biolegend.com/en-us/products/apc-cyanine7-anti-human-cd39-antibody-12925
Anti-Tim-3-BV510:
https://www.biolegend.com/en-us/products/brilliant-violet-510-anti-human-cd366-tim-3-antibody-12009
Anti-T-bet-BV711:
https://www.biolegend.com/en-us/search-results/brilliant-violet-711-anti-t-bet-antibody-7952?GroupID=BLG6433
Anti-T-Bet-BV785:
https://www.biolegend.com/en-us/products/brilliant-violet-785-anti-t-bet-antibody-15077?GroupID=BLG6433
Anti-IL-7Ra-BV711:
https://www.biolegend.com/en-us/products/brilliant-violet-711-anti-human-cd127-il-7ralpha-antibody-7947?GroupID=BLG9274
Anti-CD8-AF700:

https://www.biolegend.com/en-us/products/alexa-fluor-700-anti-human-cd8-antibody-9062?GroupID=BLG10167
Anti-CD62L-PerCP-Cy5.5
https://www.biolegend.com/en-us/products/percp-cyanine5-5-anti-human-cd62l-antibody-4243?GroupID=BLG10270
Anti-FOXO1-AF488:
https://www.cellsignal.com/products/antibody-conjugates/foxo1-c29h4-rabbit-mab-alexa-fluor-488-conjugate/58223
Anti-TCF1/TCF7-PE:
https://www.cellsignal.com/products/antibody-conjugates/tcf1-tcf7-c63d9-rabbit-mab-pe-conjugate/14456
Anti-TCF1/TCF7-AF647:
https://www.cellsignal.com/products/antibody-conjugates/tcf1-tcf7-c63d9-rabbit-mab-alexa-fluor-647-conjugate/6709
Anti-BCL-2-PE:
https://www.cellsignal.com/products/antibody-conjugates/bcl-2-124-mouse-mab-pe-conjugate/26295
Anti-LEF1-AF488:
https://www.cellsignal.com/products/antibody-conjugates/lef1-c12a5-rabbit-mab-alexa-fluor-488-conjugate/8490
Anti-LEF1-PE:
https://www.cellsignal.com/products/antibody-conjugates/lef1-c12a5-rabbit-mab-pe-conjugate/14440
Anti-PD-1-PE-Cy7:
https://www.thermofisher.com/antibody/product/CD279-PD-1-Antibody-clone-eBioJ105-J105-Monoclonal/25-2799-42
Anti-LAG3-PE:
https://www.thermofisher.com/antibody/product/CD223-LAG-3-Antibody-clone-3DS223H-Monoclonal/12-2239-42
Anti-CD45-PerCP-Cy5.5
https://www.thermofisher.com/antibody/product/CD45-Antibody-clone-HI30-Monoclonal/45-0459-42
Anti-CD62L-PE:
https://www.bdbiosciences.com/en-us/products/reagents/flow-cytometry-reagents/research-reagents/single-color-antibodies-ruo/pe-mouse-anti-human-cd62l.555544
Anti-CD271-Biotin:
https://www.biolegend.com/en-gb/products/biotin-anti-human-cd271-ngfr-antibody-17603
FOXO1 Rabbit mAB: validated on HEK293T cells:
https://www.cellsignal.com/products/primary-antibodies/foxo1-c29h4-rabbit-mab/2880
Lamin A Mouse mAb: validated on HeLa, PC-3, A549, PANC-1, MCF7, and ACHN cells:
https://www.cellsignal.com/products/primary-antibodies/lamin-a-133a2-mouse-mab/86846
GAPDH Mouse mAb: validated on HeLA, NIH/3T3, C6, and COS-7 cells:
https://www.cellsignal.com/products/primary-antibodies/gapdh-d4c6r-mouse-mab/97166
Anti-rabbit IgG, HRP-linked Antibody:
https://www.cellsignal.com/products/secondary-antibodies/anti-rabbit-igg-hrp-linked-antibody/7074
Anti-mouse IgG, HRP-linked Antibody:
https://www.cellsignal.com/products/secondary-antibodies/anti-mouse-igg-hrp-linked-antibody/7076

# Eukaryotic cell lines

Policy information about cell lines and Sex and Gender in Research

| Cell line source(s) | Nalm6 and 143B cells were obtained from the American Type Culture Collection and engineered as per methods. HEK239GP cells were obtained from the National Cancer Institute. Primary human T cells were obtained from anonymous healthy donor buffy coats via the Stanford University Blood Center under a University Institutional Review Board-exempt protocol or Human Peripheral Blood Leukopaks (StemCell Technologies) at Stanford and from the University of Pennsylvania Human Immunology Core at Children's Hospital of Philadelphia. |
|---|---|
| Authentication | Nalm6 and 143B cell lines that were engineered to express luciferase and fluorescent proteins (Nalm6-GL and 143B-GL) were verified via flow cytometry. CD19 negative Nalm6 used in tumor re-challenge experiments were verified via flow cytometry. Nalm6 and 143B cell lines and engineered versions of these cell lines were previously authenticated via STR fingerprinting prior to their use in this study. |
| Mycoplasma contamination | Cells were frequently tested for mycoplasma using the Lonza MycoAlert Mycoplasma Detection kit. All experiments reported in this study used cells that tested negative for Mycoplasma. |
| Commonly misidentified lines (See ICLAC register) | None were used. |

# Animals and other research organisms

Policy information about studies involving animals; ARRIVE guidelines recommended for reporting animal research, and Sex and Gender in Research

| Laboratory animals | NOD/SCID/IL2Rγ-/- (NSG) mice were bred, housed, and treated under Stanford University APLAC- or Children's Hospital of Philadelphia (CHOP) ACUP-approved protocols. 6-8 week-old mice were healthy, immunocompromised, drug- and test-naïve, and unused in other procedures. Mice were housed at the Stanford Veterinary Service Center (VSC) or CHOP Department of Veterinary Services (DVR) in a barrier facility with a 12-hour light/dark cycle, and mice were kept at a temperature of 20-23C (CHOP) or 20-26C (Stanford) with humidity ranging from 30-70%. |
|---|---|

| Wild animals | No wild animals were used in this study |
|---|---|
| Reporting on sex | Relatively equal numbers of male and female healthy human donor T cells were used for this study. Similarly, in vivo experiments used relatively equal numbers of male and female mice (but were sex-controlled within each individual experiment). Therefore, findings from this study can be applied to both sexes. |
| Field-collected samples | No field samples were used. |
| Ethics oversight | All animal studies were undertaken under Stanford University APLAC- or Children's Hospital of Philadelphia (CHOP) ACUP-approved protocols. Mice were monitored daily by VSC or DVR staff and euthanized if endpoint criteria were met. |

Note that full information on the approval of the study protocol must also be provided in the manuscript.

# Flow Cytometry

## Plots

Confirm that:

☒ The axis labels state the marker and fluorochrome used (e.g. CD4-FITC).

☒ The axis scales are clearly visible. Include numbers along axes only for bottom left plot of group (a 'group' is an analysis of identical markers).

☒ All plots are contour plots with outliers or pseudocolor plots.

☒ A numerical value for number of cells or percentage (with statistics) is provided.

## Methodology

| Sample preparation | For surface phenotyping: as per methods, T cells were washed twice in FACS buffer (PBS + 2% FBS), stained with fluorophore-conjugated antibodies in FACS buffer (100uL total staining volume per sample) for 30 minutes on ice, washed twice again with FACS buffer, and then analyzed. |
|---|---|
| | For intracellular phenotyping: as per methods, cells were prepared as above with surface stains then fixed, permeabilized, and stained using the eBioscience FoxP3 Transcription Factor Staining Buffer Set as per manufacturer's protocol. |
| | Cell surface antibodies were used at a 1:100 dilution during staining, with the exception anti-14g2a and anti-FMC63, which were used at a 1:1000 dilution. Intracellular antibodies were used at a 1:50 dilution and live/dead staining was used at a 1:1000 dilution. |
| Instrument | BD Fortessa (Stanford) and Cytek Aurora (Children's Hospital of Philadelphia) |
| Software | FACS Diva version 10.8.1 (Stanford) or SpectroFlo (Children's Hospital of Philadelphia) |
| Cell population abundance | For most phenotyping experiments, between 50,000-500,000 lymphocytes were collected. |
| Gating strategy | Samples were gated on lymphocytes (FSC-A/SSC-A), single cells (SSC-W/SSC-H), and relevant markers (tNGFR, CAR, CD4, CD8, etc. as specified in the manuscript main text). For cells that were stained with live/dead staining, live cells were also gated into the population of interest (live/dead staining was performed using either Zombie NIR Fixable Viability Kit [Biolegend catalog #423105], Fixable Viability Kit eFluor 506 [eBioscience catalog #65-0866-18], or Fixable Viability Kit eFluor 780 [eBioscience catalog #65-0865-14]). |
| | For FOXO1 CRISPR KO studies, cells were gated as per above and additionally FOXO1KO cells were gated on the FOXO1 negative subpopulation; AAVS1-edited controls were analyzed regardless of FOXO1 expression. |
| | For in vivo murine blood analysis, samples were gated on lymphocytes and single cells as above, GFPhi tumor cells were gated out via the FITC channel, Human CD45hi cells were gated in, and CountBright absolute counting beads were used to validate absolute cell numbers. |

☒ Tick this box to confirm that a figure exemplifying the gating strategy is provided in the Supplementary Information.

