## [Peer Review File · Nature]

Manuscript Title: FOXO1 is a master regulator of CAR T memory programming

Reviewer Comments & Author Rebuttals

Reviewer Reports on the Initial Version:

Referees' comments:

Referee #1 (Remarks to the Author):

A. Summary of the key results

FOXO1 over-expression (but not nuclear retention variants or impaired DNA-binding variants) in multiple CAR-T models in vitro and in vivo shows memory modulation via epigenetic modulation towards improved functionality and persistence. The importance of FOXO1 is further illustrated by upregulated gene expression in the FOXO1 regulon in persisting CART products in patients with durable remissions on human CART trials.

B. Originality and significance

This work is original and significant. Importantly this paper defines a mechanism for the biological effects of FOXO1 by examining the differential function of variants, underscoring the importance of understanding the biology of FOXO1 to be able to design the optimal modular cassette. To my knowledge this nuanced evaluation has not been successfully undertaken by other groups.

C. Data & methodology: validity of approach, quality of data, quality of presentation

Valid, high-quality analyses. Valuable to have replicated the dataset in haem and onc models, and in a model designed to recapitulate exhaustion. Further, excellent to note the confirmation of FOXO1 signature in trial patient products which further increases the impact here.

D. Appropriate use of statistics and treatment of uncertainties

Appropriate use of statistics

E. Conclusions: robustness, validity, reliability

Robust, valid and reliable conclusions in multiple model settings.

F. Suggested improvements: experiments, data for possible revision

None

G. References: appropriate credit to previous work?

Appropriate

H. Clarity and context: lucidity of abstract/summary, appropriateness of abstract, introduction and conclusions

Clear messaging throughout and easy to follow.

Referee #2 (Remarks to the Author):

CAR-T cell therapy has emerged as both a novel and potent form of cancer therapy. One of the hurdles to this approach is the persistence and retained function of the adoptively transferred engineered cells. In this work the authors present data suggesting that engineering the over expression of FOXO1 might enhance memory like features of CAR-T cells and thus enhance their function. Overall, this is a well executed study with robust findings. Based on the literature these findings are not terribly surprising but the bottom line is this group has expertly and definitively proven this hypothesis. Such findings have clear clinical implications.

From an immunologic perspective, the group has revealed interesting and surprising findings regarding the different functions of FOXO1 and TCF-1 both of which have been associated with T cell memory. To this end I think there are a number of questions that the authors should more thoroughly answer in order to clarify these findings.

First, what is the role of TCF-1 in the FOXO1^{oe} cells? That is, what is the gene expression profile and function of cells in which FOXO1 is overexpressed and TCF7 is deleted?

Second, What is the gene expression profile of FOXO1 nucleus-retained overexpressing cell versus the FOXO1 WT over expression cells? How do these differences account for their differences in function?

Finally, Figure 6 describes enhanced activity of the FOXO1^{oe} cells in a solid tumor model. What is the mechanism of this enhancement? Is it simply that the CAR-T persist longer. Or are these cells truly engineered to overcome the hurdles encountered which account for the decreased efficacy of CAR-T in solid tumors?

Referee #3 (Remarks to the Author):

The manuscript by Doan et al evaluates memory-associated transcription factors FOXO1 and TCF1 for reprogramming of CAR T cells to enhance memory reprogramming and antitumor activity. Authors first show that inhibition of FOXO1, using pharmacological and gene knockout strategies,

promotes the acquisition of an exhaustion-like phenotype and reduces the in vivo antitumor potential of CD19-CAR T cells. Next, authors compare the overexpression of FOXO1 and TCF1 and show that only FOXO1-OE promoted memory gene expression programs and enhanced antitumor function in both blood and solid tumor xenograft models. Based on the FOXO1-KO and OE gene expression profiles, authors define a FOXO1 'regulon' consisting of 41 putative target genes. Evaluating public datasets of clinical CAR and TIL patient outcomes, authors find a correlation between clinical response and T cell products with a high FOXO1 regulon score. A bit unexpected, TCF1-OE did not promote T cell memory and instead skewed CAR T cells to a more T_{pex}-like state.

This study adds to the current literature detailing the importance of the T cell memory state as a core determinant of CAR T cell function and provides insights into the differential roles for transcription factors FOXO1 and TCF1 in memory reprogramming. Overall, this is a strong study that is technically sound and well-presented. There are some remaining questions to be addressed which are detailed below, including a more detailed assessment the role for FOXO1 in CD4 vs CD8 T cell subsets.

Specific Comments:

1. Ext data Fig 1 shows that FOXO1 inhibition reduces CAR-T expansion and reduces numbers of CD8 T cells in the product. Do FOXO1-KO cells also exhibit reduced proliferation. It is not clear the mechanism for the reduction in CD8 cells. Is there evidence for a differential dependency on FOXO1 between CD8 vs CD4 subsets (see also comment 5).
2. In vitro killing and proliferation for FOXO1-KO CAR T cells is not shown. This would help support the mechanism by which the KO cells reduces the antitumor activity of CD19-CARs.
3. Does FOXO1-OE maintenance of a memory state negatively impact the rate of killing or cytolytic activity of CAR T cells? For recursive killing assay in Fig 2c, please provide data for tumor cell enumeration at each time point. Were technical replicates included in this assay? While the HA-*tonic* signaling model is useful for evaluating chronic CAR signaling, additional effector functional and phenotypic analysis of non-tonically signaling CAR T cells post-tumor challenge is warranted.
4. For Fig 2, what is the rationale for evaluating only CD8 CAR-T recursive killing and expansion. Was a similar difference observed for CD4 CAR T cells. Since CAR products contain both CD4 and CD8 subsets, defining the role of FOXO1 in both T cell subsets would be important.
5. The FOXO1-3a data is a bit unclear based on previously published literature and the data presented. The increase in memory marker expression for FOXO1-3a is not convincing from data presented Ext data Fig 6. Is there additional data to support increased FOXO1 activity and/or memory phenotype, for instance post tumor challenge? The lack of killing in Ext Data Fig 6e is quite striking, and how is this reconciled with the improvement in antitumor activity in Fig 5.
6. For panels quantifying and/or phenotyping CAR T cells were cells gated on both CAR and tNGFR? Can authors comment on the retention or depletion of TCF or FOXO1 expression in both in vivo and in vitro studies.
7. Is there data to support that FOXO1dbd alters transcription or chromatin remodeling function of FOXO1. For Figure 5f, please provide quantitation for the soluble fraction and chromatin bound fraction. Also please provide flow cytometry for FOXO1 levels for OE vs DBD variants.
8. FOXO1 expression/activity has been reported to alter the homing properties of lymphocytes (eg. PMID: 26789248) and authors show that FOXO1-OE maintains the expression lymphoid homing markers IL7R and CD62L on T cells. Does FOXO1-OE CAR T cells have altered tumor trafficking

properties. While in vivo efficacy presented in in Fig 6 supports tumor homing, additional data evaluating CAR trafficking would strengthen the manuscript.

Minor comments:

1. Specify in methods FOXO1 antibody used to detect KO and/or OE
2. For flow cytometry plots showing OE of FOXO1, FOXO1-3a and TCF1 it would be helpful to include histograms for the iso control. This would better show that tNGFR is still uniformly positive.
3. For extended datasets, it would be helpful in several of the flow panels to add either %pos or MFI to support conclusion in body of manuscript.
4. Confirm labeling of graphs for Fig 6g and 6h

Author Rebuttals to Initial Comments:

We would like to thank the reviewers for their insightful critiques. We have addressed each of their comments point-by-point and have revised the manuscript accordingly (highlighted text). We have added a significant amount of new data to this manuscript (over 40 new subfigures), which collectively support the overarching premise that FOXO1 is a critical regulator of memory and antitumor activity in human CAR T cells, and that FOXO1 overexpression can augment CAR T cell potency by counteracting exhaustion and increasing persistence. Our major new findings include:

- Endogenous *TCF7* is not required for FOXO1_{OE}-associated transcriptional reprogramming and enhanced antitumor activity.
- Tumor-infiltrating FOXO1_{OE} CAR T cells are enriched in tissue residence- and effector-associated gene expression programs, which are consistent with durable FOXO1_{OE} function and solid tumor control in this model.
- FOXO1 knockout and/or pharmacologic inhibition diminishes CAR T expansion and increases apoptosis, which likely limits antitumor activity.
- CAR T cells that overexpress nuclear-restricted FOXO1 (FOXO1_{3A}) are transcriptomically distinct from those that overexpress wild-type FOXO1 (FOXO1_{OE}) and are de-enriched in T cell catabolism and activation-related pathways, consistent with reduced antitumor function.
- CAR T cells overexpressing a variant of FOXO1 with lower affinity DNA-binding (FOXO1_{DBD}) exhibit perturbed transcriptional and epigenetic reprogramming compared to FOXO1_{OE}, including less accessible bZIP family motifs.

Referee #1 (Remarks to the Author):

A. Summary of the key results:

FOXO1 over-expression (but not nuclear retention variants or impaired DNA-binding variants) in multiple CAR-T models in vitro and in vivo shows memory modulation via epigenetic modulation towards improved functionality and persistence. The importance of FOXO1 is further illustrated by upregulated gene expression in the FOXO1 regulon in persisting CART products in patients with durable remissions on human CART trials.

B. Originality and significance:

This work is original and significant. Importantly this paper defines a mechanism for the biological effects of FOXO1 by examining the differential function of variants, underscoring the importance of understanding the biology of FOXO1 to be able to design the optimal modular cassette. To my knowledge this nuanced evaluation has not been successfully undertaken by other groups.

C. Data & methodology: validity of approach, quality of data, quality of presentation:

Valid, high-quality analyses. Valuable to have replicated the dataset in haem and onc models, and in a model designed to recapitulate exhaustion. Further, excellent to note the confirmation of FOXO1 signature in trial patient products which further increases the impact here.

D. Appropriate use of statistics and treatment of uncertainties:

Appropriate use of statistics

E. Conclusions: robustness, validity, reliability:

Robust, valid and reliable conclusions in multiple model settings.

F. Suggested improvements: experiments, data for possible revision:

None

G. References: appropriate credit to previous work?:

Appropriate

H. Clarity and context: lucidity of abstract/summary, appropriateness of abstract, introduction and conclusions:
Clear messaging throughout and easy to follow.

We thank the reviewer for their comments and appreciate their support of our study.

Referee #2 (Remarks to the Author):

CAR-T cell therapy has emerged as both a novel and potent form of cancer therapy. One of the hurdles to this approach is the persistence and retained function of the adoptively transferred engineered cells. In this work the authors present data suggesting that engineering the over expression of FOXO1 might enhance memory like features of CAR-T cells and thus enhance their function. Overall, this is a well executed study with robust findings. Based on the literature these findings are not terribly surprising but the bottom line is this group has expertly and definitively proven this hypothesis. Such findings have clear clinical implications.

From an immunologic perspective, the group has revealed interesting and surprising findings regarding the different functions of FOXO1 and TCF-1 both of which have been associated with T cell memory. To this end I think there are a number of questions that the authors should more thoroughly answer in order to clarify these findings.

First, what is the role of TCF-1 in the FOXO1^{oe} cells? That is, what is the gene expression profile and function of cells in which FOXO1 is overexpressed and TCF7 is deleted?

We thank the reviewer for their thoughtful assessment of our manuscript. To address this point, we used CRISPR/Cas9 gene-editing to disrupt the expression of endogenous *TCF7* in FOXO1^{OE} CAR T cells targeting CD19. Using RNA-seq, we show that *TCF7* knockout does not dramatically alter the transcriptome of FOXO1^{OE} CAR T cells *in vitro* (New Fig.5f-g, New Extended Data Fig. 8f), indicating that FOXO1^{OE}-mediated transcriptional reprogramming occurs independently of endogenous *TCF7*.

We next infused *TCF7* gene-edited CAR T cells into leukemia-bearing mice. Consistent with transcriptomic analyses, *TCF7* knockout did not affect the enhanced antitumor function of FOXO1^{OE} CAR T cells *in vivo* (New Fig. 5h). Collectively, these data indicate that transcriptional reprogramming and enhanced function endowed by FOXO1^{OE} is independent of endogenous *TCF7* and further support the notion that FOXO1, rather than TCF1, is the central regulator of memory and stemness in human CAR T cells. New text elaborating on these findings was added in lines 317-323

Second, what is the gene expression profile of FOXO1 nucleus-retained overexpressing cell versus the FOXO1 WT over expression cells? How do these differences account for their differences in function?

To assess the gene expression profile of nuclear-restricted FOXO1^{3A} CAR T cells, we performed new bulk RNA-seq analyses. FOXO1^{3A} cells displayed an altered transcriptome that was enriched in pathways related to non-coding RNA processing (ncRNA) and ribosome biogenesis and de-enriched in those related to catabolic processes and T cell activation (New Extended Data Fig. 6f,g). This transcriptional profile is consistent with functional data showing blunted killing, cytokine secretion, proliferation, and tumor control in FOXO1^{3A} CAR T cells (Fig. 5a-e and New Extended Data Fig. 6h,i). While the biological significance of ncRNA processing and ribosome biogenesis in this model system is unclear, it's worth noting that FOXO1^{KO} cells were de-enriched in ncRNA processing and ribosome

biogenesis (New Extended Data Fig. 2f). These data suggest that ncRNA processing and ribosome biogenesis are processes that are promoted by FOXO1, and that FOXO1_{3A} enforces these processes beyond FOXO1_{OE} levels. Future studies are warranted to determine whether ncRNA processing and ribosome biogenesis are mechanistically important for memory T cell function.

New biochemical studies confirm that FOXO1_{3A} indeed preferentially localizes to the nucleus and is bound to chromatin (New Extended Data Fig. 6b), and is therefore expected to constitutively enforce FOXO1-related processes compared to FOXO1_{OE}. In summary, we posit that nuclear-restricted FOXO1 partially “locks” CAR T cells into a memory-like state and opposes their activation and differentiation into potent effector T cells. Conversely, wild-type FOXO1_{OE} can freely translocate between the nucleus and cytosol and therefore enables cell state plasticity (i.e. both memory-like and effector-like function). New text describing these experiments is included in lines 278-285.

Finally, Figure 6 describes enhanced activity of the FOXO1_{oe} cells in a solid tumor model. What is the mechanism of this enhancement? Is it simply that the CAR-T persist longer? Or are these cells truly engineered to overcome the hurdles encountered which account for the decreased efficacy of CAR-T in solid tumors?

We agree with the reviewer’s comment and therefore sought to identify the mechanism(s) by which FOXO1_{OE} enhances CAR T activity against solid tumors. We repeated our 143B osteosarcoma xenograft model and performed single-cell RNA-sequencing on tumor-infiltrating Her2.BBz CAR T cells (New Fig 6f-i, New Extended Data Fig. 9). Transcriptomic analyses demonstrate that FOXO1_{OE} CAR T cells exhibit enriched effector gene expression signature (New Fig 6i and New Extended Data Fig. 9g) compared to control tNGFR CAR T cells, which is consistent with enhanced cytokine secretion *ex vivo* (New Fig. 6e). We also detected higher expression of a canonical tissue residence signature (New Fig. 6i and New Extended Data Fig. 9h), suggesting that these cells may display enhanced tumor retention. Consistent with this notion, we observed higher total numbers of tumor-infiltrating FOXO1_{OE} CAR T cells compared to control tNGFR CAR T cells (New Fig. 6c).

Tumor-infiltrating FOXO1_{OE} CAR T did not display a canonical memory-like phenotype; however, they were enriched with a FOXO1 transcriptomic signature that was derived from bulk RNA-seq (New Fig. 6i), suggesting that exogenous FOXO1 activity is in fact retained within the TME and that FOXO1_{OE} cells retain the plasticity needed to differentiate into potent effector T cells. Finally, flow cytometry data from our first submission showing reduced exhaustion markers and increased CD62L in splenic FOXO1_{OE} cells were not reproduced in follow-up experiments and may be attributed to donor T cell variability. We therefore excluded these data from our revised manuscript and instead centered our conclusions on new scRNA-seq data.

Collectively, these data suggest that FOXO1_{OE} CAR T cells control solid tumor growth *in vivo* via enhanced persistence, maintenance of effector function, and tumor residence. New text describing these findings has been added to lines 330-341.

Referee #3 (Remarks to the Author):

The manuscript by Doan et al evaluates memory-associated transcription factors FOXO1 and TCF1 for reprogramming of CAR T cells to enhance memory reprogramming and antitumor activity. Authors first show that inhibition of FOXO1, using pharmacological and gene knockout strategies, promotes the acquisition of an exhaustion-like phenotype and reduces the *in vivo* antitumor potential of CD19-CAR T cells. Next, authors compare the overexpression of FOXO1 and TCF1 and show that only FOXO1-OE promoted memory gene expression programs and enhanced antitumor function in both blood and solid tumor xenograft models. Based on the FOXO1-KO and OE gene expression profiles,

authors define a FOXO1 'regulon' consisting of 41 putative target genes. Evaluating public datasets of clinical CAR and TIL patient outcomes, authors find a correlation between clinical response and T cell products with a high FOXO1 regulon score. A bit unexpected, TCF1-OE did not promote T cell memory and instead skewed CAR T cells to a more T_{pex}-like state.

This study adds to the current literature detailing the importance of the T cell memory state as a core determinant of CAR T cell function and provides insights into the differential roles for transcription factors FOXO1 and TCF1 in memory reprogramming. Overall, this is a strong study that is technically sound and well-presented. There are some remaining questions to be addressed which are detailed below, including a more detailed assessment the role for FOXO1 in CD4 vs CD8 T cell subsets.

We thank the reviewer for their helpful comments, which we believe greatly improved the quality of our manuscript.

Specific Comments:

1. Ext data Fig 1 shows that FOXO1 inhibition reduces CAR-T expansion and reduces numbers of CD8 T cells in the product. Do FOXO1-KO cells also exhibit reduced proliferation? It is not clear the mechanism for the reduction in CD8 cells. Is there evidence for a differential dependency on FOXO1 between CD8 vs CD4 subsets (see also comment 5).

We repeated FOXO1_{KO} experiments with 3 new healthy donors and observed a modest but statistically significant reduction in *in vitro* expansion (New Extended Data Fig. 2a). However, cell trace violet and annexin V experiments did not demonstrate consistent differences in FOXO1_{KO} doublings post-tumor challenge or viability, respectively (Response to Reviewers Fig. 1, below). For annexin V experiments, our results may be explained by technical limitations associated with co-detection of annexin V and intracellular FOXO1 (i.e., apoptotic or dead cells exhibit reduced FOXO1 expression regardless of knockout status).

Redacted

Since treatment with a small molecule inhibitor of FOXO1 (FOXO1_i) phenocopies FOXO1_{KO} and similarly blunts CAR T expansion, we instead assayed apoptosis during pharmacologic FOXO1 inhibition. FOXO1_i cells exhibited a reduction in live cells and a concomitant increase in necrotic cells at Day 15 (New Extended Data Fig. 1d), a timepoint at which FOXO1 is active and promotes expression of memory markers (Fig. 1d,e). These data are consistent with a model wherein FOXO1 activity is required for maintaining memory-like CAR T cell viability and that increased apoptosis is likely one mechanism limiting FOXO1_{KO} CAR T cell persistence.

Additionally, we generated new phenotyping data which show that both CD4⁺ and CD8⁺ FOXO1_{KO} CAR T cells have diminished expression of memory markers IL-7R α , CD62L, and TCF1 (New Fig. 1d,e and New Extended Data Fig. 2b). However, only CD8⁺ FOXO1_{KO} CAR T cells exhibit concomitant increases in the terminal effector markers CD39 and T-bet (New Fig. 1d and New Extended Data Fig. 2b). These phenotypic data and the defect in FOXO1_{KO} CD8⁺ expansion suggest that CD8⁺ cells may be more sensitive to terminal differentiation and/or exhaustion following FOXO1_{KO} relative to CD4⁺ cells. Repeat phenotyping experiments did not reproduce increased expression of TIM-3 upon FOXO1_{KO}; therefore, these data were removed from the revised manuscript. New text detailing our findings was added to the manuscript in lines 154-156 and 413-416.

2. In vitro killing and proliferation for FOXO1-KO CAR T cells is not shown. This would help support the mechanism by which the KO cells reduces the antitumor activity of CD19-CARs.

In addition to the proliferation and apoptosis assays described in our response to comment #1, we also performed new *in vitro* killing assays using CD19.BBz FOXO1_{KO} CAR T cells against Nalm6 leukemia. We did not observe any significant differences in cytotoxicity *in vitro* (Response to Reviewers Fig 2, below). We posit that the relatively short 2-4 day *in vitro* killing assay may not provide adequate resolution to detect manifestation of exhaustion and/or loss of CAR T persistence, both of which limit antitumor activity *in vivo*. Future studies are warranted to fully elucidate the mechanisms by which endogenous FOXO1 affects antitumor activity.

Redacted

3. Does FOXO1-OE maintenance of a memory state negatively impact the rate of killing or cytolytic activity of CAR T cells? For recursive killing assay in Fig 2c, please provide data for tumor cell enumeration at each time point. Were technical replicates included in this assay? While the HA-tonic signaling model is useful for evaluating chronic CAR signaling, additional effector functional and phenotypic analysis of non-tonically signaling CAR T cells post-tumor challenge is warranted.

Data included in the original manuscript demonstrate that FOXO1_{OE} CAR T cells exhibit an enhanced ability to cytolyse, proliferate, secrete cytokine, and control tumor growth *in vivo* compared to control CAR T (Figs. 5 and 6 and Extended Data Fig. 7). These results, in addition to new scRNA-seq showing enhanced effector gene expression in tumor-infiltrating FOXO1_{OE} CAR T cells (New Fig. 6f-i), suggest that FOXO1_{OE} CAR T exhibit the plasticity needed to differentiate from memory-like cells into potent effector cells. We posit that endogenous post-translational regulation of FOXO1 subcellular localization (i.e. nuclear vs cytosolic) is the mechanism by which FOXO1_{OE} CAR T can toggle between memory and effector states (New Extended Data Fig. 6b). Text relating to these results is included in the Discussion in lines 423-435.

The CD19.28 ζ serial restimulation assay in Fig. 2c was performed with an E:T ratio that ensured complete tumor clearance in all conditions; therefore, tumor cell enumeration in this assay is not possible. 2 biological replicates were performed in this experiment and are displayed as “donor 1” and “donor 2” in Fig. 2c. To directly assess tumor killing, Incucyte assays were performed with HA.28 ζ CAR T cells, which demonstrate that FOXO1_{OE} enhances killing in settings of chronic stimulation (Extended Data Fig. 3f).

We agree that assessment of non-tonic signaling CAR T cell models are warranted. To clarify, the functional and phenotypic data for serially restimulated CD19.28 ζ CAR T cells (referenced in the second paragraph of our response to this comment) is shown in Fig. 2c-f and Extended Data Fig. 3c-d, while data for tonic signaling HA.28 ζ CAR T cells are shown in Figs. 2g-k and Extended Data Fig 3e-f. Both tonic- and non-tonic signaling CAR models support the conclusion that that FOXO1_{OE} enhances CAR T antitumor function in settings of chronic stimulation.

4. For Fig 2, what is the rationale for evaluating only CD8 CAR-T recursive killing and expansion. Was a similar difference observed for CD4 CAR T cells. Since CAR products contain both CD4 and CD8 subsets, defining the role of FOXO1 in both T cell subsets would be important.

We agree with the reviewer on the importance of defining the role of FOXO1_{OE} in CD4 and CD8 subsets. The rationale for showing CD8s in Fig. 2C is due to the selective effect of FOXO1_{OE} on CD8 expansion *in vitro*, which also manifested *in vivo* (Fig. 5d and New Fig. 6d). Absolute CD4 CAR T expansion was not significantly impacted by FOXO1_{OE} *in vitro*, but was augmented in leukemia and osteosarcoma xenograft models (New Extended Figs. 7d and 9b).

To further define the role of FOXO1_{OE} in CD4 T cells, we performed bulk RNA-sequencing on *in vitro* CAR T cells expressing the tonic signaling HA.28 ζ CAR (New Extended Data Fig. 4). We detected thousands of DEGs in FOXO1_{OE} vs tNGFR CD4⁺ CAR T cells, many of which were also differentially regulated in CD8s. These include upregulation of memory-associated genes (*IL7R*, *LEF1*, *KLF3*) and downregulation of exhaustion-associated genes (*TOX*, *CD244*). Principal component analysis shows that FOXO1_{OE} samples cluster together regardless of CD4 or CD8 identity, indicating similar FOXO1-specific reprogramming in both CD4s and CD8s (New Extended Data Fig. 4b). New *in vivo* scRNA-seq on tumor-infiltrating FOXO1_{OE} CAR T cells demonstrate that CD8s (New Fig. 6i), and CD4s to a lesser extent (not shown), are enriched in T cell effector gene signatures. Despite literature implicating FOXO1 in the formation and maintenance of Tregs, FOXO1 overexpression had negligible effects on T_{REG} formation *in vitro* and *in vivo* compared to tNGFR controls (New Extended Data Figs. 4e and 9f,i).

Collectively, these data and those described in response to Comment #1 suggest that endogenous FOXO1 and FOXO1_{OE} similarly affect the phenotype and function of CD4⁺ CAR T cells in short-term assays, but their effects on expansion, persistence, and effector function are more pronounced for CD8⁺ CAR T cells.

5. The FOXO1-3a data is a bit unclear based on previously published literature and the data presented. The increase in memory marker expression for FOXO1-3a is not convincing from data presented Ext data Fig 6. Is there additional data to support increased FOXO1 activity and/or memory phenotype, for instance post tumor challenge? The lack of killing in Ext Data Fig 6e is quite striking, and how is this reconciled with the improvement in antitumor activity in Fig 5.

We apologize for the lack of clarity around FOXO1_{3A} and do not claim increased memory marker expression in FOXO1_{3A} compared to FOXO1_{OE}. Extended Data Fig. 6c-e are intended to demonstrate

that FOXO1_{OE} and FOXO1_{3A} express similar levels of memory markers. We have further clarified this point in the manuscript text.

Since FOXO1_{3A} is nuclear-restricted (and should therefore function as a more active variant), we performed additional experiments to assess FOXO1 chromatin binding. During transient activation, T cells expressing FOXO1_{3A} exhibit a higher ratio of chromatin-bound to cytosolic FOXO1 (New Extended Data Fig. 6b), consistent with resistance to nuclear export and increased FOXO1 activity. We posit that this could be one mechanism by which FOXO1_{3A} blunts effector function *in vitro*.

We also assessed the gene expression profile of nuclear-restricted FOXO1_{3A} CAR T cells using bulk RNA-seq (which is described in greater detail in our response to reviewer 2 comment #2). Briefly, FOXO1_{3A} cells displayed an altered transcriptome that was enriched in pathways related to non-coding RNA processing and ribosome biogenesis and de-enriched in those related to catabolic processes and T cell activation relative to FOXO1_{OE} (New Extended Data Fig. 6f,g). This transcriptional profile is consistent with functional data showing blunted killing, cytokine secretion, proliferation, and tumor control in FOXO1_{3A} CAR T cells compared to FOXO1_{OE} (Fig. 5a-e and Extended Data Fig. 6h,i).

The reviewer correctly points out a modest enhancement in antitumor activity in leukemia-bearing mice treated with FOXO1_{3A} CD19.28z CAR T cells (Fig. 5a-e), which is inconsistent with the striking lack of killing observed *in vitro* (Extended Data Fig. 6h). We posit that constitutive expression of FOXO1_{3A} delays, but does not fully ablate, the ability of CAR T cells to mount effector responses, and that its effect may be dose-dependent. Indeed, other studies have demonstrated that low and/or transient FOXO1_{3A} expression can induce a memory phenotype without blunting T cell effector function^{21,62,63}. We discuss the role of FOXO1_{3A} expression levels and kinetics in the Discussion in lines 423-435.

Overall, these studies suggest that overexpression of nuclear-restricted FOXO1_{3A} partially opposes CAR T cell differentiation and effector function. Conversely, wild-type FOXO1_{OE} endows a degree of cell state plasticity and dramatically enhances CAR T effector function.

6. For panels quantifying and/or phenotyping CAR T cells were cells gated on both CAR and tNGFR? Can authors comment on the retention or depletion of TCF or FOXO1 expression in both *in vivo* and *in vitro* studies.

In some phenotyping experiments, cells were gated on both CAR and tNGFR (ex. Fig. 2e). Our gating strategies for FOXO1_{KO} and FOXO1_{OE} experiments are clarified in New Supplemental Data Figure 1. In experiments that required an additional channel for FACS phenotyping, we used magnetic beads to preselect on tNGFR⁺ cells and then gated on CAR⁺ cells during flow cytometry analyses (ex. Fig. 2f).

FOXO1 and TCF1 expression were maintained during the *in vitro* expansion period, as evidenced by detection of TF protein (Fig. 2b). In leukemia xenograft models, we detected surface tNGFR on TCF1_{OE} and FOXO1_{OE} cells 3-weeks post-infusion, indicating durable expression of these transgenes (Extended Fig. 7e). Finally, new scRNA-seq data demonstrate that tumor-infiltrating FOXO1_{OE} CAR T cells were enriched in a FOXO1_{OE} transcriptomic signature derived from bulk RNA-seq studies (Fig. 3 and New Fig. 6i), suggesting that exogenous FOXO1 remains active in the tumor microenvironment approximately 6 weeks post-transduction.

7. Is there data to support that FOXO1dbd alters transcription or chromatin remodeling function of FOXO1. For Figure 5f, please provide quantitation for the soluble fraction and chromatin bound fraction. Also please provide flow cytometry for FOXO1 levels for OE vs DBD variants.

New bulk RNA-seq and bulk ATAC-seq data demonstrate that FOXO1_{DBD} does in fact alter the transcriptional and epigenetic programming promoted by wild-type FOXO1_{OE} (New Extended Data Fig. 8c,d).

Briefly, both FOXO1_{OE} and FOXO1_{DBD} cells displayed increased accessibility of forkhead box family motifs; but FOXO1_{DBD} lacked the concomitant increases in accessibility of bZIP family motifs that were detected in FOXO1_{OE}. At the transcriptomic level, only 50% of the DEGs in FOXO1_{OE} were shared with FOXO1_{DBD} when compared to tNGFR, indicating perturbed exogenous FOXO1 transcriptional activity, which is consistent with the observed defect in chromatin binding (Extended Data Fig. 8a) and antitumor activity (Extended Data Fig. 8e). Collectively, these data suggest that FOXO1_{OE} DNA binding is required for increased CAR T potency *in vivo*. New text describing these findings are included in lines 308-315 and 425-427.

8. FOXO1 expression/activity has been reported to alter the homing properties of lymphocytes (e.g. PMID: 26789248) and authors show that FOXO1-OE maintains the expression lymphoid homing markers IL7R and CD62L on T cells. Does FOXO1-OE CAR T cells have altered tumor trafficking properties. While *in vivo* efficacy presented in in Fig 6 supports tumor homing, additional data evaluating CAR trafficking would strengthen the manuscript.

We refer the reviewer to our response to Reviewer 2 Comment #3. Briefly, we performed additional solid tumor xenograft experiments to assess Her2.BB ζ CAR T trafficking to the spleen and the tumor. We detected significantly higher numbers of FOXO1_{OE} CAR T in both the tumor and the spleen compared to tNGFR CAR T cells (New Fig. 6c and New Extended Data Fig. 9a). New scRNA-seq data demonstrate increased expression of core tissue residence genes in FOXO1_{OE} CAR T compared to tNGFR CAR T (New Fig. 6i), which is consistent with enhanced tumor localization and tumor control in FOXO1_{OE}-treated mice.

To further interrogate the trafficking capacity of FOXO1_{OE} CAR T cells, we also performed *in vitro* chemotaxis assays using human Her2.BB ζ cells. We tested CAR T cells derived from 3 healthy human donors against a CCL19 gradient (lymphoid homing chemokine) or a CXCL11 gradient (tumor homing chemokine). We did not detect statistically significant differences between FOXO1_{OE} and tNGFR CAR T in these assays (Response to Reviewers Fig. 3, below).

Redacted

We posit that fully murine tumor models would likely provide a more physiologic system to interrogate FOXO1_{OE} T cell trafficking; however, these experiments are outside the scope of our study, which is focused specifically on human CAR T cell biology and mechanisms of persistence/exhaustion. Nevertheless, our new *in vivo* data strongly implicate enhanced tumor trafficking and/or tumor residence in FOXO1_{OE} CAR T cells (lines 333-336).

Minor comments:

1. Specify in methods FOXO1 antibody used to detect KO and/or OE

This information is now included in lines 610-611.

2. For flow cytometry plots showing OE of FOXO1, FOXO1-3a and TCF1 it would be helpful to include histograms for the iso control. This would better show that tNGFR is still uniformly positive.

This figure has now been updated to include isotype controls.

3. For extended datasets, it would be helpful in several of the flow panels to add either %pos or MFI to support conclusions in body of manuscript.

Histograms in Extended Data Figs. 1e, 2g, 3a, 6c-d, and 8b have been updated to include MFIs.

4. Confirm labeling of graphs for Fig 6g and 6h

Thank you for noticing this error. Labeling has now been corrected.

Reviewer Reports on the First Revision:

Referees' comments:

Referee #1 (Remarks to the Author):

A. Summary of the key results

- As per my prior review. The new data adds weight to the manuscript, particularly the TCF1 independence and the interrogation of impact in CD4 vs CD8 T-cells.

B. Originality and significance: if not novel, please include reference

- Novel, mechanistic study which is likely to influence adoptive cellular therapy design in the clinic in the near future.

C. Data & methodology: validity of approach, quality of data, quality of presentation

- Valid, high-quality data.

D. Appropriate use of statistics and treatment of uncertainties

- Appropriate

E. Conclusions: robustness, validity, reliability

- Robust conclusions in multiple extended analyses

F. Suggested improvements: experiments, data for possible revision

- None

G. References: appropriate credit to previous work?

- Agree

H. Clarity and context: lucidity of abstract/summary, appropriateness of abstract, introduction and conclusion

- Clear, intelligible manuscript

Referee #2 (Remarks to the Author):

The authors have addressed my comments.

Referee #3 (Remarks to the Author):

The author's provide a thorough response to prior comments and the revised manuscript has added significant new data to better elucidate the mechanism for FOXO1 vs TCF1 in directing T cell memory programs and augmenting antitumor activity. Overall, the authors have addressed all my prior comments. Please note, some of the figure panels were difficult to read. For instance, for Extended Fig 1e the MFI is difficult to read due to size and color of font. Figure 3c, 4e and eFig5e are also difficult to read due to the small font size. For Fig 6, it is suggested to specify in the legend that T cells were infused by TVI.